# Image Editing As Programs with Diffusion Models

**Yujia Hu[1], Songhua Liu[2,1], Zhenxiong Tan[1], Xingyi Yang[3,1], and Xinchao Wang[1]** *

[1]National University of Singapore
[2]School of Artificial Intelligence, Shanghai Jiao Tong University
[3]The Hong Kong Polytechnic University
`yujia.hu@u.nus.edu, xinchao@nus.edu.sg`

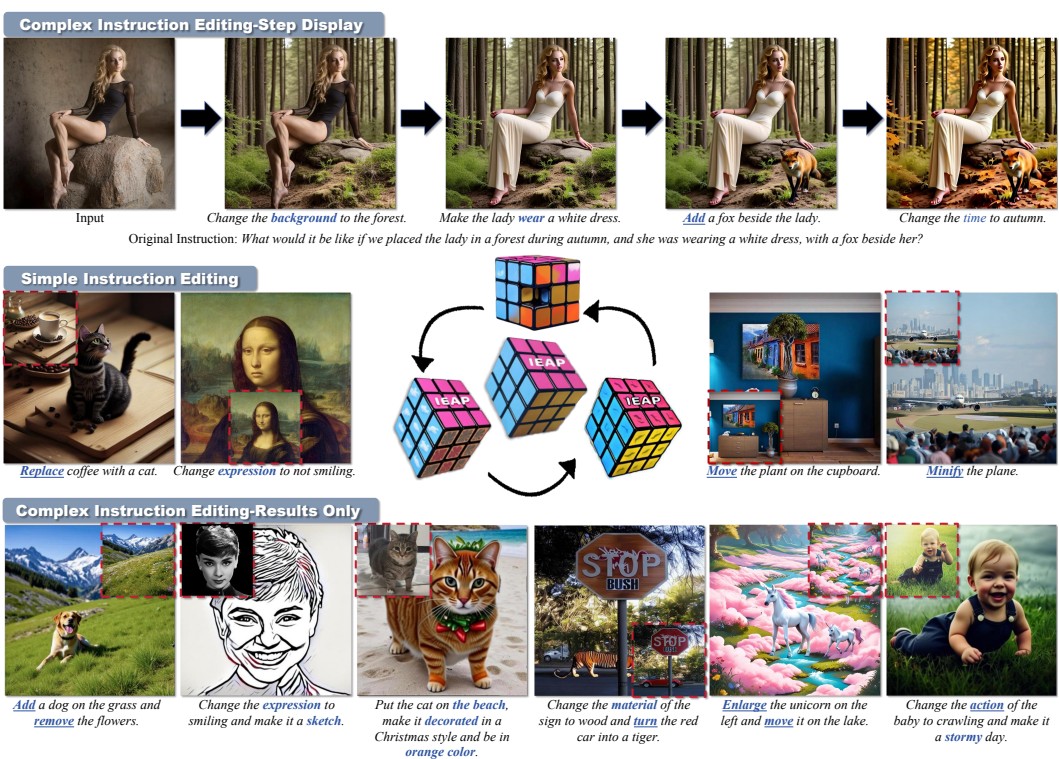

Figure 1: Visual results of our IEAP. Rows 1 and 3 showcase complex multi-step edits (Row 1 is further decomposed into individual instructions), while Row 2 shows single-instruction edits. Single instructions are underlined if needing to be reduced to atomic operations.

## Abstract

While diffusion models have achieved remarkable success in text-to-image generation, they encounter significant challenges with instruction-driven image editing. Our research highlights a key challenge: these models particularly struggle with structurally-inconsistent edits that involve substantial layout changes. To address this gap, we introduce *Image Editing As Programs* (IEAP), a unified image editing framework built upon the Diffusion Transformer (DiT) architecture. Specifically, IEAP deals with complex instructions by decomposing them into a sequence of programmable *atomic* operations. Each atomic operation manages a specific type of structurally consistent edit; when sequentially combined, IEAP enables the execution of arbitrary and structurally-inconsistent transformations. This re-

---

*Corresponding Author

39th Conference on Neural Information Processing Systems (NeurIPS 2025).

ductionist approach enables IEAP to robustly handle a wide spectrum of edits, encompassing both structurally-consistent and inconsistent changes. Extensive experiments demonstrate that IEAP significantly outperforms state-of-the-art methods on standard benchmarks across various editing scenarios. In these evaluations, our framework delivers superior accuracy and semantic fidelity, particularly for complex, multi-step instructions. Codes are available here.

# 1 Introduction

Image editing lies at the heart of a wide range of applications from photo retouching and content creation to visual storytelling and scientific visualization [42, 5, 63]. With the advent of diffusion models [23, 53, 47], the field has shifted towards highly precise and controllable manipulations [45, 12, 62]. The inherently progressive denoising process enables multi-stage pipelines [24, 4, 2] and localized editing methods [10, 77, 58], and its native support for multi-modal inputs has inspired unified frameworks that integrate heterogeneous signals within a single model [33, 15, 71, 18].

More recently, text-to-image pipelines based on Diffusion Transformers (DiTs) [46, 13, 31] have set new standards in generative fidelity. However, their capacity for instruction-driven editing [41, 27] remains under-explored. Notably, although there are a few existing methods [80, 37] that have extended DiTs to instruction-driven editing, they are always restricted to a narrow set of common editing operations and lack evaluation on comprehensive editing tasks.

To address this limitation, we initiate a taxonomy study of image editing instructions to systematically assess the editing capabilities of current DiT-based conditional generation methods. Our empirical analysis reveals an interesting performance dichotomy: While current methods demonstrate proficiency in structurally-consistent edits where the layouts of the input and output images remain aligned, they exhibit significant degradation when handling structurally-inconsistent operations that require layout modifications.

To overcome this issue, we introduce ***Image Editing As Programs*** (IEAP), a unified framework atop the DiT architecture which is capable of handling diverse types of editing operations efficiently and robustly in this paper. Notably, we show that structurally-inconsistent instructions can in fact be reduced to a small set of simple operations, which are called as *atomic* operations in our paper. Thus, instead of treating each edit as a monolithic, end-to-end task, IEAP leverages the Chain-of-Thought (CoT) reasoning [65] to break the original editing command into a sequence of atomic operations, which are namely Region of Interest (RoI) localization, RoI inpainting, RoI editing, RoI compositing and global transformation, and then executes them in a sequential manner via a neural program interpreter [49].

The five atomic operations serve as the fundamental building blocks for complex editing tasks. As such, through the sequential combination of atomic operations, IEAP can robustly handle complex, multi-step instructions that are typically confound in conventional end-to-end approaches.

Extensive experiments show that our framework demonstrates state-of-the-art performance across standard benchmarks, excelling in both structural preservation and alteration tasks through atomic-level operation decomposition compared to other approaches. Simultaneously, the CoT reasoning and programming pipeline of IEAP enable significantly more accurate and semantically more coherent edits under complex, multi-step instructions even compared to the leading proprietary models.

Our main contributions can be summarized as follows:

- We present a comprehensive taxonomy and empirical analysis of instruction-driven editing in DiT-based conditional generation, revealing a performance dichotomy between structurally-consistent and -inconsistent edits.

- We introduce ***Image Editing As Programs*** (IEAP), a unified framework on the DiT backbone that leverages CoT reasoning to parse free-form instructions into sequential atomic operations and then executes them sequentially by a neural program interpreter, thereby enabling robust handling of layout-altering and complex edits.

- Extensive experiments demonstrate that IEAP achieves state-of-the-art performance in both structure-preserving and -altering scenarios, delivering notably higher accuracy and semantic fidelity especially on complex, multi-step instructions compared to existing methods.

## 2 Related Work

**Conditional image generation.** Early conditional image generation approaches like ControlNet [77] typically adopt plug-in control adapters to incorporate single condition [3, 16, 35] like segmentation mask or diverse conditional inputs [82, 48, 26, 40, 70] to guide the generation of images. Recently, the field of conditional image generation has witnessed remarkable breakthroughs through the integration of DiTs [13, 46, 31], with continuous innovations improving output quality and edit precision [45]. Some methods [69, 32, 68, 9] aim to create a unified DiT foundation for versatile conditional image generation and editing by integrating diverse inputs within a single framework. while approaches like OminiControl [60] and so on [61, 79, 38, 80, 67] leverage LoRA-based fine-tuning [25] for lightweight and effective control.

**Instructional image editing.** Instruction-based image editing [41, 27] enables intuitive, language-driven modifications of existing images. Early works like InstructPix2Pix [6] establishes paired instruction–image datasets for supervised fine-tuning of diffusion models. For subsequent works, some of them focus on architectural refinement [38, 37, 81, 34, 20], which introduce specialized conditioning units and multi-stage training to improve control granularity and consistency, others concentrate on data-centric enhancements [76, 17, 56, 8], that expand instruction coverage and diversify edit examples. Moreover, some approaches [75, 28, 33, 15] has unified LLM-based [1] language reasoning with diffusion-based synthesis in a single framework, and some [72, 78] leverage CoT [65] and in-context learning [21] to enhance the reasoning ability of models for more complex editing tasks. Meanwhile, certain approaches [64, 66, 54] build upon multimodal understanding to decompose intricate instructions, ensuring editing precision and output reliability. More recently, some works [14, 80, 37] have advanced image editing with DiTs. For instance, ICEdit [80] leverages the in-context generation capabilities of large-scale DiTs to achieve flexible few-shot instruction editing, while Step1X-Edit [37] focuses on large-scale data construction and multi-modal integration to enable general-purpose image editing with performance approaching proprietary models.

## 3 Motivation

### 3.1 Preliminaries

**Diffusion Transformer Fundamentals.** The image generation process of text-guided DiTs [46, 13, 31] is accomplished by successively denoising input tokens in multiple steps. At step $t$, the model processes:

$$\mathbf{S}_t = [\mathbf{X}_t, \mathbf{C}_T] \tag{1}$$

where $\mathbf{X}_t \in \mathbb{R}^{N \times d}$ represents noisy image tokens and $\mathbf{C}_T \in \mathbb{R}^{M \times d}$ denotes text tokens, they share the embedding dimension $d$. Image tokens use Rotary Position Embedding (RoPE) [59] with spatial coordinates $(i, j)$, while text tokens fix positions at $(0, 0)$, enabling Multi-Modal Attention (MMA) [44] mechanisms to model cross-modal interactions.

**Unified Conditioning Framework.** To integrate visual control signals, the prior work [60] extends the baseline formulation by incorporating encoded condition images:

$$\mathbf{S}_t = [\mathbf{X}_t, \mathbf{C}_T, \mathbf{C}_I] \tag{2}$$

where $\mathbf{C}_I \in \mathbb{R}^{N \times d}$ denotes latent tokens from condition images via the pretrained VAE encoder [30, 52]. This unified sequence enables tri-modal fusion within transformer architectures, eliminating spatial misalignment inherent in feature concatenation baselines.

Moreover, an auxiliary adaptive positional encoding mechanism further preserves spatial consistency across these modalities by assigning coordinates to each token type with minimal overhead.

**Gap in Instruction-Driven DiT Editing.** Despite the rapid advances in DiT-based conditional image generation [60, 79, 38, 67], research on instruction-driven editing [41, 27] remains scarce. The few existing methods [80, 37] that do support instructional edits are typically confined to a small set of routine operations, and lack a comprehensive evaluation across diverse editing scenarios, leaving DiT's true editing potential unclear. This gap motivates us to conduct a taxonomy study of DiT's ability in instructional image editing, which is detailed in Sec. 3.2.

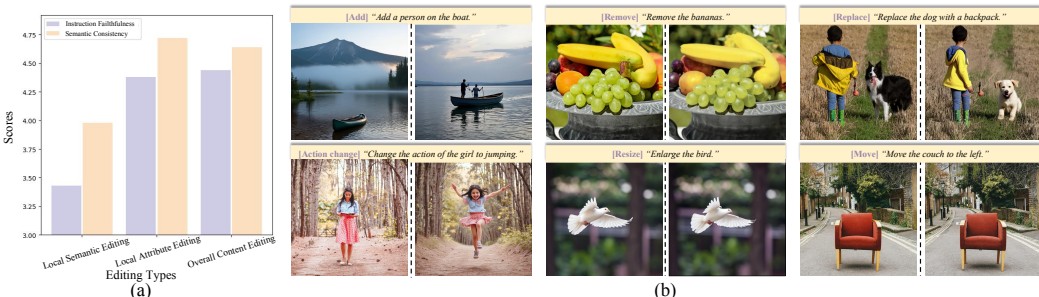

Figure 2: Results of our preliminary experiments. Figure (a) shows the GPT-4o scores for three editing types across instruction faithfulness and semantic consistency, ranging from 1 to 5. Figure (b) shows the representative failure cases from local semantic editing.

## 3.2 Preliminary Experiments and Observations

To this end, we conduct a comprehensive evaluation of diffusion models for instruction-driven editing, uncovering an interesting performance dichotomy: *While these methods excel at structurally-consistent edits, they falter dramatically on structurally-inconsistent operations that demand explicit layout modifications.*

**Taxonomy and Experimental Setup.** To enable systematic analysis [27, 73, 72], we first categorize instruction-based image editing into three main types: local semantic editing, which modifies the identity, position or size, e.g., add, remove, replace, action change, move and resize; local attribute editing, which adjusts certain properties of objects, e.g., color change, texture change, appearance change, expression change, and background change; and overall content editing, which alters the whole image consistently, e.g., tone transfer and style change.

Then we use AnyEdit dataset [73] and OminiControl [60] to train models on the above editing types, accompanied by GPT-4o [29] to rate each edit on instruction faithfulness and semantic consistency.

**Results and Analysis.** As shown in Fig. 2(a), both local attribute editing and overall content editing attain relatively high GPT-4o scores, whereas local semantic editing exhibits a notable performance drop. As illustrated in Fig. 2(b), the cases of "add" and "action change" alter unrelated areas like the background, and the remaining four cases demonstrate a complete failure.

We attribute this discrepancy to the fact that, unlike local attribute and overall content edits, local semantic edits require explicit spatial-layout modifications. For instance, "add" and "delete" operations necessitate instance-level scene recomposition, while "move" and "resize" further demand precise coordinate system recalibration.

**Key Insight.** Based on the above analysis, spatial-layout modification remains a critical challenge for diffusion-based editing models; conversely, edits that preserve the original layout demonstrate substantially better performance. We speculate that, with limited training data, it is difficult for the model to learn the complex patterns underlying layout-changing tasks. Although DiT architectures [46, 13, 31] employ powerful full-attention mechanisms to capture long-range dependencies, they still struggle with editing operations that require nontrivial scene reconfiguration.

Due to the combinatorial complexity of spatial-layout modifications and the empirical limitations of DiT architectures, we propose to simplify the layout-editing paradigm through decomposition, which is detailed in Sec. 4.

## 4 Methods

### 4.1 Program with Atomic Operations

The insight in Sec. 3.2 motivates us to decouple semantic and spatial reasoning. Building on this foundation, we propose a programmatic reduction framework that systematically decomposes complex editing instructions into modular atomic operations. Specifically, we first formulate instruction-driven image editing as an executable program via Chain-of-Thought (CoT) reasoning [65], and then use a neural program interpreter [49] to transcode the reasoning graph into a dynamic execution plan, sequentially invoking relevant atomic modules.

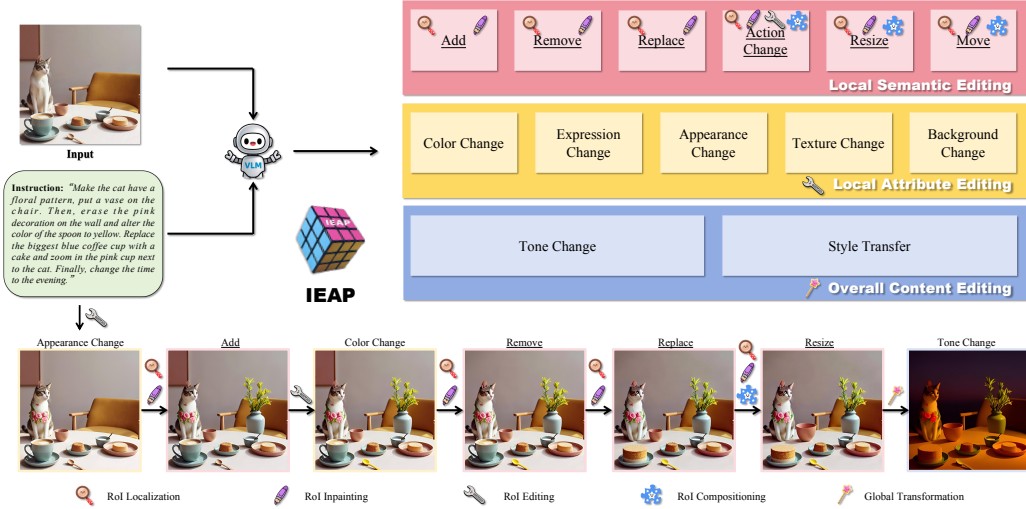

Figure 3: Our pipeline. The original instruction is first parsed by a VLM into atomic operations, which are then sequentially executed via a neural program interpreter.

## 4.2 General Pipeline

We abstract all editing instructions into five atomic primitives: (1) RoI Localization: Identify and isolate the relevant region in the image that the instruction refers to, serving as the spatial grounding step for subsequent localized edits; (2) RoI Inpainting: Introduce new visual content or remove existing elements within the localized region, enabling semantic-level additions, substitutions, or deletions; (3) RoI Editing: Modify visual attributes within the region, such as color, texture, or appearance, to reflect fine-grained property changes specified by the instruction; (4) RoI Compositing: Reintegrate the edited region into the full image while preserving spatial coherence and visual continuity; (5) Global Transformation: Adjust the overall content for coherent full-image modifications, such as changing the illumination, weather, or style of the whole image.

The overall pipeline is shown as Fig. 3. We reduce any editing instruction into an arbitrary combination of the five atomic operations described above, which can be formulated as:

$$T \equiv \bigoplus_{k=1}^{K} \mathcal{A}_k, \quad \mathcal{A}_k \in \{\mathcal{A}_{\text{loc}}, \mathcal{A}_{\text{inp}}, \mathcal{A}_{\text{edit}}, \mathcal{A}_{\text{comp}}, \mathcal{A}_{\text{global}}\} \quad (3)$$

where $T$ denotes the free-form editing instruction, $\bigoplus$ represents the sequential program combination, $K$ is the number of atomic operations, $\mathcal{A}_{\text{loc}}$, $\mathcal{A}_{\text{inp}}$, $\mathcal{A}_{\text{edit}}$, $\mathcal{A}_{\text{comp}}$, and $\mathcal{A}_{\text{global}}$ represent the five atomic primitives respectively.

**RoI Localization.** All problematic local semantic edits share a common first step: localizing a Region of Interest (RoI) in the image for editing. Given an image $I$ and an editing instruction $T$, we first employ a Large Language Model (LLM) [1] to locate the text RoI:

$$\rho = M_{\text{LLM}}(T), \quad (4)$$

where $\rho$ represents the text RoI extracted by the LLM $M_{\text{LLM}}$. Subsequently, we achieve accurate localization of image RoI by:

$$R = M_{\text{seg}}(I, \rho), \quad (5)$$

where $R$ denotes the image RoI segmented by the segmentation model $M_{\text{seg}}$ [74].

For add operation, the instruction may not specify a text RoI, or the specification may be ambiguous. In such cases, we first derive the overall layout of all candidate objects using the capability of segmentation models [50, 74], and then prompt the LLM to determine the appropriate image RoI based on $T$.

Regarding move and resize, once the image RoI is obtained, we update the spatial layout of the image using an LLM [1]. Specifically, we provide the LLM with a set of in-context examples that define our layout representation and demonstrate representative editing patterns [36]. Given the current layout $L$ and the instruction $T$, the LLM is prompted to produce a modified layout $L_{\text{edit}}$, as formulated below:

$$\text{Tags} = M_{\text{LLM}}(I), \quad L = M_{\text{seg}}(\text{Tags}), \quad L_{\text{edit}} = M_{\text{LLM}}(L, T). \quad (6)$$

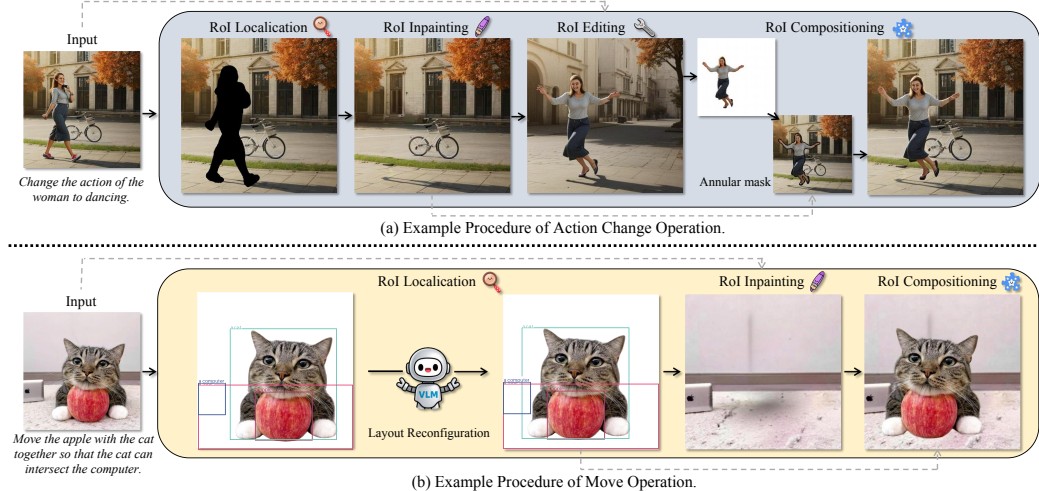

Figure 4: Example procedure. Figure (a) and Figure (b) illustrate the procedures of action change and movement respectively.

We then derive the geometric differences between $L$ and $L_{\text{edit}}$ and convert them into the corresponding affine transformations, consisting of translation, scaling, and reshaping, and apply it to $R$ to update the spatial configuration, yielding the transformed mask $R'$.

**RoI Inpainting.** Once the image RoI has been localized, we apply inpainting to seamlessly fill and complete the region. For additive and substitutive operations, which aim to introduce new objects, we employ a prompt-conditioned inpainting process to guide the generation of new content. Specifically, we first extract the semantic entity $E$ from the instruction $T$ via an LLM [1]:

$$E = M_{\text{LLM}}(T), \tag{7}$$

and then construct a composite prompt $P$ in the form: *"add E on the black region"*. For removal operations, which aim to eliminate existing content without introducing new semantics, we adopt a background-oriented infilling strategy, setting $P$ as *"fill in the hole of the image"*. The edited image $I_{\text{edit}}$ is then generated by:

$$I_{\text{edit}} = M_{\text{inpaint}}\left(I \odot (1 - R), P\right), \tag{8}$$

where $M_{\text{inpaint}}$ denotes the inpainting model trained by us.

**RoI Editing.** When operations pertain to property change are performed, we use the trained attribute editing model $M_{\text{attr}}$ to perform edits in this stage to obtain $I_{\text{edit}}$:

$$I_{\text{edit}} = M_{\text{attr}}\left(I, T\right). \tag{9}$$

**RoI Compositing.** To ensure seamless integration of the edited RoI with its surrounding context, we first construct an annular mask $M_{\text{ann}}$ by applying morphological dilation and erosion [51, 55] to the transformed RoI mask $R'$:

$$M_{\text{ann}} = \text{Dilate}(R', k_1) \setminus \text{Erode}(R', k_2). \tag{10}$$

Then, we employ a fusion network $M_{\text{fusion}}$, trained on ring-masked object boundaries, to refine the pre-composited image $I_{\text{prep}}$ using the generated annular mask. The final edited image is obtained as:

$$I_{\text{edit}} = M_{\text{fusion}}\left(I_{\text{prep}} \odot (1 - M_{\text{ann}}), P\right), \tag{11}$$

where $P$ is set as "inpaint the black-bordered region so that the object's edges blend smoothly with the background" to guide seamless boundary blending.

**Global Transformation.** Like RoI editing, in the scenarios involving global transformation, we use the trained global transformation model $M_{\text{global}}$ to perform edits in this final stage to obtain $I_{\text{edit}}$.

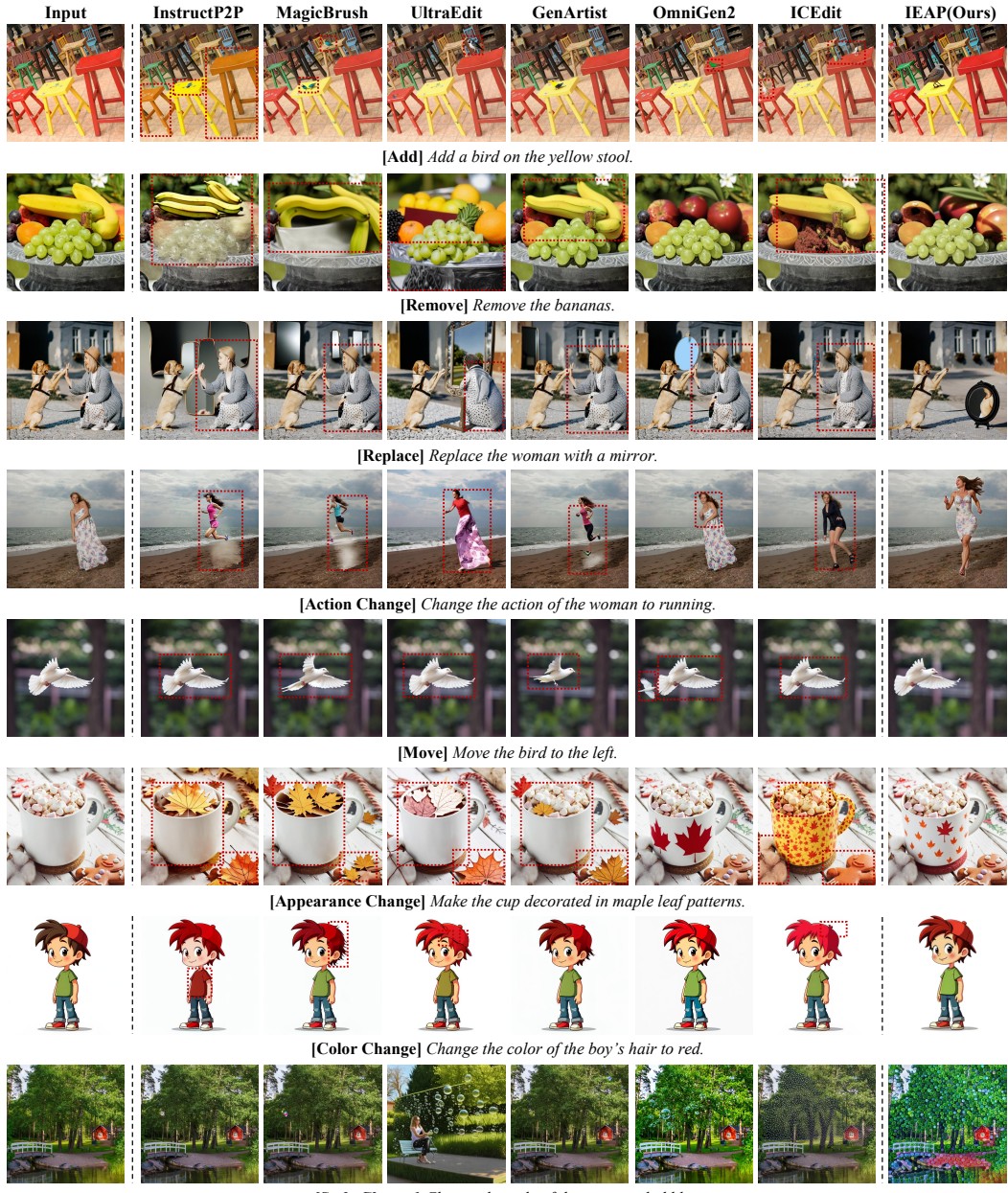

Figure 5: Comparison results of ours with baseline methods on representative editing cases. Others exhibit poor performance even on some common editing operations, while our approach demonstrates superior effectiveness across all operations.

# 5 Experiments

## 5.1 Experimental Settings

**Training Settings.** We train four specialized models for RoI inpainting, RoI editing, RoI compositing, and global transformation respectively. All models are fine-tuned on FLUX.1-dev [31] using LoRA [25], with default settings for rank 128 and alpha 128. Training is conducted with a batch size of 1 and runs for 50,000 iterations each. We use the Prodigy optimizer [39], enabling safeguard warmup and bias correction, with a weight decay of 0.01. The experiments are conducted on single NVIDIA H100 GPU (80GB).

| Method | MagicBrush test | | | | AnyEdit test | | | |
|---|---|---|---|---|---|---|---|---|
| | CLIP$_{im}$ ↑ | CLIP$_{out}$ ↑ | L1 ↓ | DINO ↑ | CLIP$_{im}$ ↑ | L1 ↓ | DINO ↑ | GPT ↑ |
| InstructPix2Pix | 0.838 | 0.229 | 0.112 | 0.758 | 0.801 | 0.110 | 0.765 | 3.83 |
| MagicBrush | 0.886 | 0.241 | 0.074 | 0.859 | 0.824 | 0.128 | 0.742 | 3.90 |
| UltraEdit | 0.911 | 0.227 | 0.061 | 0.883 | 0.833 | 0.114 | 0.772 | 3.93 |
| GenArtist | 0.908 | 0.232 | 0.063 | 0.889 | 0.829 | 0.121 | 0.776 | 3.98 |
| OmniGen2 | 0.881 | 0.242 | 0.100 | 0.830 | 0.857 | 0.132 | 0.772 | 4.13 |
| ICEdit | 0.913 | 0.236 | **0.058** | 0.885 | 0.847 | 0.110 | 0.765 | 4.13 |
| Ours | **0.922** | **0.247** | 0.060 | **0.897** | **0.882** | **0.096** | **0.825** | **4.41** |

Table 1: Quantitative results on MagicBrush and AnyEdit test set.

| Method | Local Semantic Editing | | | | Local Attribute Editing | | | | Overall Content Editing | | | |
|---|---|---|---|---|---|---|---|---|---|---|---|---|
| | CLIP$_{im}$ ↑ | L1 ↓ | DINO ↑ | GPT ↑ | CLIP$_{im}$ ↑ | L1 ↓ | DINO ↑ | GPT ↑ | CLIP$_{im}$ ↑ | L1 ↓ | DINO ↑ | GPT ↑ |
| InstructP2P | 0.826 | 0.132 | 0.738 | 3.74 | 0.790 | 0.135 | 0.737 | 3.92 | 0.766 | 0.156 | 0.642 | 3.91 |
| MagicBrush | 0.860 | 0.106 | 0.796 | 3.90 | 0.809 | 0.117 | 0.762 | 4.21 | 0.763 | 0.187 | 0.616 | 3.99 |
| UltraEdit | 0.867 | 0.095 | 0.812 | 3.86 | 0.801 | 0.092 | 0.793 | 3.94 | 0.754 | 0.201 | 0.611 | 4.41 |
| GenArtist | 0.864 | 0.097 | 0.821 | 3.88 | 0.814 | 0.108 | 0.801 | 3.96 | 0.752 | 0.207 | 0.595 | 4.38 |
| OmniGen2 | 0.893 | 0.097 | 0.834 | 4.12 | 0.832 | 0.114 | 0.786 | 4.06 | 0.783 | 0.224 | 0.634 | 4.42 |
| ICEdit | 0.881 | 0.088 | 0.810 | 4.08 | 0.825 | 0.095 | 0.795 | 4.06 | 0.759 | 0.188 | 0.603 | 4.45 |
| Ours | **0.907** | **0.081** | **0.854** | **4.42** | **0.861** | **0.083** | **0.821** | **4.54** | **0.895** | **0.107** | **0.879** | **4.51** |

Table 2: Quantitative results on different types of editing operations.

**Dataset Setup.** For both the RoI editing and global transformation models, we sample from the relevant subsets of the AnyEdit [73] dataset and apply GPT-4o [29] to filter the data of some types that have numerous noisy examples. To cover facial expression edits absent in AnyEdit, we integrate the CelebHQ-FM dataset [11], which offers consistent identities and annotated expressions suitable for our instruction schema. The RoI inpainting and RoI compositing models are trained on samples from the "add", "remove" and "replace" splits of AnyEdit. For each sample, we first obtain the image RoI according to the editing instruction. In the RoI Inpainting training setup, we set the pixels within image RoI to black as input to train. For RoI Compositing, we set $k_1$ and $k_2$ as 3 in default to blackout the annular mask region of image RoI as input for training.

**Evaluation Settings.** We evaluate our method on two benchmarks: MagicBrush test set [76], a widely used dataset spanning diverse editing types, and AnyEdit test set [73], from which we select 16 instruction-based editing categories. For MagicBrush, we follow previous works [76, 81, 15, 56] and report CLIPimg, CLIPout [22], $L_1$, and DINO [7, 43] scores to measure the similarity between the generated results and ground-truth images. While for AnyEdit, where some categories lack reference captions required for calculating CLIPout, we instead leverage GPT-4o [29] to rate each edited image on a scale from 1 to 5 across three dimensions: instruction faithfulness, semantic consistency, and aesthetic quality, with the final GPT score obtained by averaging the three aspect scores.

We first compare our method with existing state-of-the-art open-source baselines, including Instruct-Pix2Pix [6], MagicBrush [76], UltraEdit [81], GenArtist [64], OmniGen2 [66] and ICEdit [80]. In addition, to demonstrate the competitiveness of our approach against powerful proprietary multimodal foundation models in complex image editing scenarios, we further make comparisons with SeedEdit (Doubao) [57], Gemini 2.0 Flash [19], and GPT-4o [29].

## 5.2 Comparisons with State of the Art.

**Qualitative Comparisons.** Fig. 5 shows the results of our approach against other six methods [6, 76, 81, 64, 66, 80] on some representative editing cases. Unlike previous methods, which sometimes misinterpret or fail to execute the given instructions, modify unintended regions, introduce undesired artifacts, or produce visually implausible results, our method consistently exhibits clear and consistent advantages in accurately following the instruction, maintaining structural coherence, preserving instance-level fidelity and retaining fine-grained visual details.

**Quantitative Comparisons.** Table 1 exhibits the quantitative comparison results of our method and other approaches [6, 76, 81, 64, 66, 80] on MagicBrush test set [76] and AnyEdit test set [73]. The results show that our method demonstrates state-of-the-art performance on both datasets. On

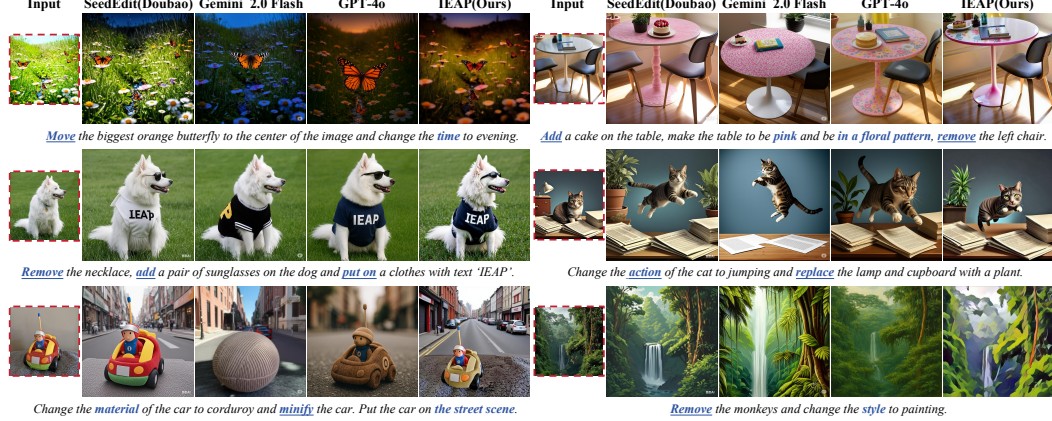

| Input | SeedEdit(Doubao) | Gemini 2.0 Flash | GPT-4o | IEAP(Ours) | Input | SeedEdit(Doubao) | Gemini 2.0 Flash | GPT-4o | IEAP(Ours) |

*Move the biggest orange butterfly to the center of the image and change the time to evening.*     *Add a cake on the table, make the table to be pink and be in a floral pattern, remove the left chair.*

*Remove the necklace, add a pair of sunglasses on the dog and put on a clothes with text 'IEAP'.*     *Change the action of the cat to jumping and replace the lamp and cupboard with a plant.*

*Change the material of the car to corduroy and minify the car. Put the car on the street scene.*     *Remove the monkeys and change the style to painting.*

Figure 6: Comparisons on Complex Instructions with Leading Multimodal Models. Our method achieves comparable or even better edit completeness and pre-post consistency.

MagicBrush, our method achieves the best performance in terms of caption alignment, semantic consistency, and preservation of fine-grained structural details. Although it incurs a marginal increase in pixel-level deviation compared to the best [80], this is far outweighed by the substantial gains in perceptual quality and semantic fidelity. Furthermore, on AnyEdit, our approach yields significant and comprehensive improvements across all evaluation metrics, further highlighting its superiority over existing techniques.

To provide a more fine-grained analysis of editing performance, we group a subset of the instruction-based categories from the AnyEdit test set [73] into three macro-tasks: local semantic editing, local attribute editing and overall semantic editing. For local attribute editing, we augment with some CelebHQ-FM [11] test images to evaluate facial expression changes. The quantitave comparison results are shown in Tab. 2, where our method consistently outperforms other candidates across all three task categories and evaluation metrics.

**Comparisons with Cutting-Edge Multimodal Models.** To demonstrate the superiority of our reduction strategy on complex editing tasks, we also conduct comparative experiments against prominent closed-source multimodal models [57, 19, 29]. As illustrated in Fig. 6, our method rivals, and in most cases surpasses the performance of these leading models on intricate scenarios requiring multiple sequential edits. Unlike competing approaches, which frequently omit specified instructions or introduce extraneous alterations unrelated to the editing directives, our framework faithfully executes each instruction while ensuring superior image consistency and instance preservation.

## 5.3 Ablation Studies

| Settings | $CLIP_{im}$ ↑ | $CLIP_{out}$ ↑ | L1 ↓ | DINO ↑ | GPT ↑ |
|---|---|---|---|---|---|
| w/o CoT & Reduction | 0.873 | 0.241 | 0.117 | 0.795 | 4.10 |
| w/o RoI Inpainting | 0.861 | 0.218 | 0.124 | 0.775 | 3.65 |
| w/o RoI Editing | 0.900 | 0.244 | 0.088 | 0.843 | 4.23 |
| w/o Layout Reconfiguration | 0.900 | 0.245 | 0.088 | 0.848 | 4.31 |
| w/o Annular Mask Integration | 0.906 | **0.252** | 0.083 | **0.854** | 4.39 |
| Full | **0.907** | **0.252** | **0.081** | **0.854** | **4.42** |

Table 3: Module-wise ablation results on AnyEdit local semantic editing test set.

Figure 7: Qualitative ablation of action change operation.

**Module-wise Ablation Studies.** To quantify the impact of each key component in our framework, we perform a series of ablation studies on the AnyEdit [73] local semantic editing test set as we split in Sec. 5.2. As shown in Tab. 3, we first substitute our CoT reasoning and reduction pipeline with end-to-end editing pipeline, resulting in a marked performance deterioration across all metrics. Next, we replace our specialized RoI inpainting and RoI editing models respectively with the generic inpainting model from [60], which induces performance declines of varying degrees. We then remove the LLM-guided layout reconfiguration and instead employing random layout modifications for relevant operations, which incurs a noticeable performance decline. Finally, omitting the annular

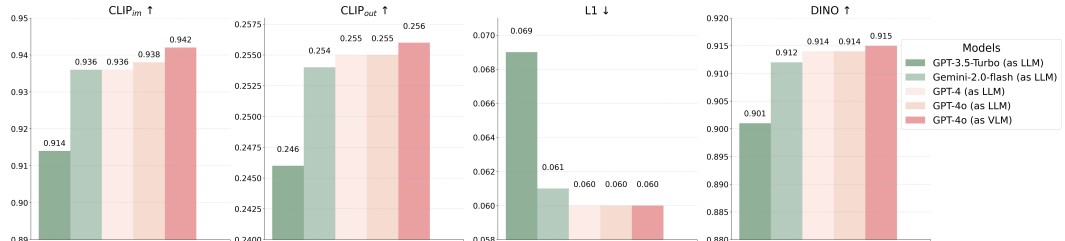

Figure 8: Ablation results of LLMs/VLMs on layout modification tasks.

mask integration produces a modest drop, underscoring its role in precise boundary delineation. Fig. 7 exhibits the ablation results on an example of "action change", visually showcasing each module's necessity. Collectively, these ablation results confirm that each component in our pipeline contributes significantly in handling robust local semantic editing tasks requiring layout changes.

**Ablation Studies on Different LLMs/VLMs.** We present comparative experimental results involving various LLMs and VLMs to analyze their influence on editing quality. For complex instruction decomposition, we evaluate challenging instructions from the MagicBrush dataset [76]. As shown in Table 4, the step remains robust across different VLMs, but performance drops notably when the original image is excluded.

| Model Type | Model | Accuracy (%) |
|---|---|---|
| VLM | GPT-4o | 100.0 |
| VLM | Gemini-2.0-flash | 96.7 |
| LLM | GPT-4 | 90.0 |
| LLM | GPT-3.5-turbo | 76.7 |

Table 4: Ablation results on complex instruction decomposition.

As in IEAP, LLMs also assist in layout modifications for "add", "move" and "resize" operations, we also compare the quantitative editing performance using different models. As shown in Fig. 8, the layout modification capacity is robust across various LLMs/VLMs generally.

## 5.4 Applications

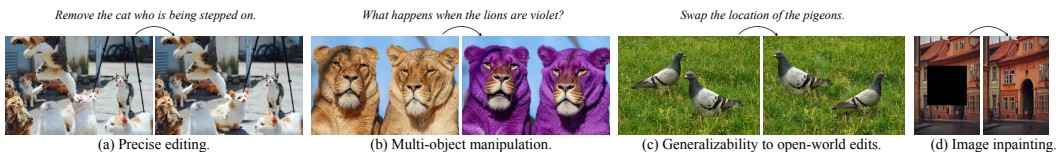

(a) Precise editing.   (b) Multi-object manipulation.   (c) Generalizability to open-world edits.   (d) Image inpainting.

Figure 9: Applications of IEAP: Handling precise, multi-object, open-world, and inpainting tasks.

Our method also demonstrates strong versatility across diverse image editing scenarios. As shown in Fig. 9(a), our method can accurately localize and edit complex target entities described in the instruction. In Fig. 9(b), when multiple synonymous objects are mentioned, all relevant instances are consistently modified. Moreover, as illustrated in Fig. 9(c), our model generalizes well to some unseen and challenging tasks such as swapping object locations, which remain difficult even for advanced large models. Finally, as shown in Fig. 9(d), our approach can naturally extend to image inpainting with satisfactory performance.

## 6 Conclusions, Limitations and Future Work

In this paper, we propose Image Editing As Programs (IEAP), a unified DiT-based framework for instruction-driven image editing. By defining five atomic operations and using CoT reasoning to convert instructions into sequential programs, IEAP processes the ability to handle both simple and complex edits. Experiments demonstrate that IEAP outperforms state-of-the-art methods in both structure-preserving and structure-altering tasks, especially for complex edits.

Despite its strong overall performance, there are also some limitations. First, for complex shadow changes, our method sometimes leaves shadows inconsistent after compositing operations. Second, multiple editing iterations may induce progressive image quality decay. Future work could focus on addressing these issues via physics-aware shadow modeling and diffusion-based quality restoration.

## Acknowledgment

This project is supported by the Ministry of Education, Singapore, under its Academic Research Fund Tier 2 (Award Number: MOE-T2EP20122-0006).

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

# Technical Appendices and Supplementary Material

In this part, we provide additional algorithm illustration, implementation details, more comparison results, more visualization results, and more analysis and discussions of the proposed approach.

## A    Algorithm Illustration

To better elaborate the details of the proposed IEAP, we provide an algorithmic illustration for the whole pipeline in Alg. 1.

---

**Algorithm 1** IEAP: Image Editing As Programs

---

**Input:**

- $I$: input image path
- $T$: original instruction
- {`RoI_Localization`, `RoI_Inpainting`, ..., `Global_Transformation`}: editing primitives
- `cot_with_gpt`($\cdot$): CoT prompt to GPT–4o
- `extract_instructions`($\cdot$): parse CoT output
- `infer_with_DiT`(op, $\cdot$): invoke DiT for primitive op
- `roi_localization`($I, instr$): returns mask for region of interest
- `fusion`($I_1, I_2$): blends two intermediate outputs
- `layout_change`($I, instr$): compute geometric transform

**Output:** final edited image $I^*$

1:  $uri \leftarrow$ `encode_image_to_datauri`$(I)$
2:  $(\mathcal{C}, \mathcal{T}) \leftarrow$ `cot_with_gpt`$(uri, T)$                     ▷ Categories and instructions
3:  $I^{(0)} \leftarrow I$
4:  **for** $i = 1$ to $|\mathcal{C}|$ **do**
5:      $cat \leftarrow \mathcal{C}[i], \quad instr \leftarrow \mathcal{T}[i]$
6:      **if** $cat \in \{\text{Add}, \text{Remove}, \text{Replace}\}$ **then**
7:          $M \leftarrow$ `roi_localization`$(I^{(i-1)}, instr)$
8:          $I' \leftarrow$ `infer_with_DiT`(RoI Inpainting, $M, instr$)
9:          $I^{(i)} \leftarrow I'$
10:     **else if** $cat = \text{Action Change}$ **then**
11:         $M \leftarrow$ `roi_localization`$(I^{(i-1)}, instr)$
12:         $I_{bg} \leftarrow$ `infer_with_DiT`(RoI Inpainting, $M, instr$)
13:         $I_{act} \leftarrow$ `infer_with_DiT`(RoI Editing, $I^{(i-1)}, instr$)
14:         $I^{(i)} \leftarrow$ `infer_with_DiT`(RoI Compositing, `fusion`$(I_{bg}, I_{act}), instr$)
15:     **else if** $cat \in \{\text{Move}, \text{Resize}\}$ **then**
16:         $M \leftarrow$ `roi_localization`$(I^{(i-1)}, instr)$
17:         $I_{bg} \leftarrow$ `infer_with_DiT`(RoI Inpainting, $M, instr$)
18:         $I_{lc} \leftarrow$ `layout_change`$(I^{(i-1)}, instr)$
19:         $I^{(i)} \leftarrow$ `infer_with_DiT`(RoI Compositing, `fusion`$(I_{bg}, I_{lc}), instr$)
20:     **else if** $cat \in \{\text{Appearance Change}, \text{Background Change},$
21:     $\text{Color Change}, \text{Material Change}, \text{Expression Change}\}$ **then**
22:         $I^{(i)} \leftarrow$ `infer_with_DiT`(RoI Editing, $I^{(i-1)}, instr$)
23:     **else if** $cat \in \{\text{Tone Transfer}, \text{Style Change}\}$ **then**
24:         $I^{(i)} \leftarrow$ `infer_with_DiT`(Global Transformation, $I^{(i-1)}, instr$)
25:     **else**
26:         **raise** ValueError("Invalid category: "$cat$")
27:     **end if**
28: **end for**
29: **return** $I^{(|\mathcal{C}|)}$

---

## B  Implementation Details

In this section, we present the prompts employed to leverage a VLM for CoT reasoning over complex instructions, providing further details on the layout-adjustment prompts.

Below are the detailed prompts used to invoke the VLM for the CoT process on complex instructions:

---

Now you are an expert in image editing. Based on the given single image, what atomic image editing instructions should be if the user wants to {instruction}? Let's think step by step.
Atomic instructions include 13 categories as follows:
- Add: Introduce a new object, person, or element into the image, e.g.: add a car on the road
- Remove: Eliminate an existing object or element from the image, e.g.: remove the sofa in the image
- Color Change: Modify the color of a specific object, e.g.: change the color of the shoes to blue
- Material Change: Alter the surface material or texture of an object, e.g.: change the material of the sign like stone
- Action Change: Modify the pose or action of an instance, e.g.: change the action of the boy to raising hands
- Expression Change: Adjust the facial expression, e.g.: change the expression to smiling
- Replace: Substitute one object in the image with a different object, e.g.: replace the coffee with an apple
- Background Change: Change the background scene to another, e.g.: change the background into forest
- Appearance Change: Modify visual attributes such as patterns or accessories, e.g.: make the cup have a floral pattern
- Move: Change the spatial position of an object within the image, e.g.: move the plane to the left
- Resize: Adjust the scale or size of an object, e.g.: enlarge the clock
- Tone Transfer: Change the global atmosphere or lighting conditions, e.g.: change the weather to foggy
- Style Change: Modify the entire image to adopt a different visual style, e.g.: make the style of the image to cartoon
Respond *only* with a numbered list. Each line must begin with the category in square brackets, then the instruction. Please strictly follow the atomic categories. The operation (what) and the target (to what) are crystal clear. Do not split replace to add and remove. Always place [Tone Transfer] and [Style Change] instructions at the end of the list.
For example:
1. [Add] add a car on the road
2. [Color Change] change the color of the shoes to blue
3. [Move] move the lamp to the left
Do not include any extra text, explanations, JSON or markdown, just the list.

---

Below are the detailed prompts used to adjust the layout of move and resize operations:

---

You are an intelligent bounding box editor. I will provide you with the current bounding boxes and the editing instruction. Your task is to generate the new bounding boxes after editing. Let's think step by step.
The images are of size 512x512. The top-left corner has coordinate [0, 0]. The bottom-right corner has coordinnate [512, 512]. The bounding boxes should not overlap or go beyond the image boundaries. Each bounding box should be in the format of (object name, [top-left x coordinate, top-left y coordinate, bottom-right x coordinate, bottom-right y coordinate]).
Do not add new objects or delete any object provided in the bounding boxes. Do not change the size or the shape of any object unless the instruction requires so.
Please consider the semantic information of the layout. When resizing, keep the bottom-left corner fixed by default. When swaping locations, change according to the center point.
If needed, you can make reasonable guesses. Please refer to the examples below:
Input bounding boxes: [("bed", [50, 300, 450, 450]), ("pillow", [200, 200, 300, 230])]
Editing instruction: Move the pillow to the left side of the bed.
Output bounding boxes: [("bed", [50, 300, 450, 450]), ("pillow", [70, 270, 170, 300])]

---

Input bounding boxes: [('a green car', [21, 281, 232, 440]), ('a blue truck', [269, 283, 478, 443]), ('a red air balloon', [66, 8, 211, 143]), ('a bird', [296, 42, 439, 142])]
Editing instruction: Move the car to the right.
Output bounding boxes: [('a green car', [81, 281, 292, 440]), ('a blue truck', [269, 283, 478, 443]), ('a red air balloon', [66, 8, 211, 143]), ('a bird', [296, 42, 439, 142])]
Input bounding boxes: [("sofa", [100, 300, 400, 400]), ("dog", [150, 250, 250, 300])]
Editing instruction: Enlarge the dog.
Output bounding boxes: [("sofa", [100, 300, 400, 400]), ("dog", [150, 225, 300, 300])]
Input bounding boxes: [("chair", [100, 350, 200, 450]), ("lamp", [300, 200, 360, 300])]
Editing instruction: Swap the location of the chair and the lamp.
Output bounding boxes: [("chair", [280, 200, 380, 300]), ("lamp", [120, 350, 180, 450])]
Now, the current bounding boxes is {bbox}, the instruction is {instruction}.

Below are the detailed prompts used to adjust the layout of add operations:

You are an intelligent bounding box editor. I will provide you with the current bounding boxes and an add editing instruction. Your task is to determine the new bounding box of the added object. Let's think step by step.
The images are of size 512x512. The top-left corner has coordinate [0, 0]. The bottom-right corner has coordinnate [512, 512].
The bounding boxes should not go beyond the image boundaries. The new box must be at least as large as needed to encompass the object. Each bounding box should be in the format of (object name, [top-left x coordinate, top-left y coordinate, bottom-right x coordinate, bottom-right y coordinate]). Do not delete any object provided in the bounding boxes. Please consider the semantic information of the layout, preserve semantic relations.
If needed, you can make reasonable guesses. Please refer to the examples below:
Input bounding boxes: [('a green car', [21, 281, 232, 440])]
Editing instruction: Add a bird on the green car.
Output bounding boxes: [('a bird', [80, 150, 180, 281]), ('a green car', [21, 281, 232, 440])]
Input bounding boxes: [('stool', [300, 350, 380, 450])]
Editing instruction: Add a cat to the left of the stool.
Output bounding boxes: [('a cat', [180, 300, 300, 450])]
Input bounding boxes: [('the white cat', [200, 300, 320, 420])]
Editing instruction: Add a hat on the white cat.
Output bounding boxes: [('the white hat', [200, 260, 320, 310]), ('cat', [200, 300, 320, 420])]
Now, the current bounding boxes is {bbox}, the instruction is {instruction}.

## C   More Quantitative Results

| Method | $\text{CLIP}_{im}\uparrow$ | $\text{CLIP}_{out}\uparrow$ | $\text{L1}\downarrow$ | $\text{DINO}\uparrow$ | $\text{GPT}_{IF}\uparrow$ | $\text{GPT}_{FC}\uparrow$ | $\text{GPT}_{AQ}\uparrow$ | $\text{GPT}_{avg}\uparrow$ |
|---|---|---|---|---|---|---|---|---|
| InstructPix2Pix | 0.847 | 0.264 | 0.092 | 0.829 | 4.50 | 4.40 | 4.26 | 4.39 |
| MagicBrush | 0.889 | 0.277 | 0.068 | 0.892 | 4.66 | 4.76 | 4.62 | 4.68 |
| UltraEdit | 0.897 | 0.274 | **0.056** | 0.909 | 3.36 | 4.24 | 4.22 | 3.94 |
| ICEdit | 0.925 | 0.277 | 0.057 | 0.915 | 4.60 | 4.80 | **4.76** | **4.72** |
| IEAP(Ours) | **0.928** | **0.278** | 0.056 | **0.917** | **4.68** | **4.84** | 4.60 | 4.71 |

Table 5: Quantitative comparison results on AnyEdit Add test set.

| Method | $\text{CLIP}_{im}\uparrow$ | $\text{CLIP}_{out}\uparrow$ | $\text{L1}\downarrow$ | $\text{DINO}\uparrow$ | $\text{GPT}_{IF}\uparrow$ | $\text{GPT}_{FC}\uparrow$ | $\text{GPT}_{AQ}\uparrow$ | $\text{GPT}_{avg}\uparrow$ |
|---|---|---|---|---|---|---|---|---|
| InstructPix2Pix | 0.800 | 0.202 | 0.108 | 0.721 | 2.74 | 3.42 | 3.20 | 3.12 |
| MagicBrush | 0.853 | 0.211 | 0.083 | 0.800 | 3.08 | 3.60 | 3.18 | 3.29 |
| UltraEdit | 0.846 | 0.211 | 0.066 | 0.802 | 2.50 | 3.54 | 3.44 | 3.16 |
| ICEdit | 0.895 | 0.212 | **0.054** | 0.875 | 4.06 | **4.48** | **4.32** | **4.29** |
| IEAP(Ours) | **0.916** | **0.230** | 0.057 | **0.886** | **4.18** | 3.88 | 3.66 | 3.91 |

Table 6: Quantitative comparison results on AnyEdit Remove test set.

| Method | CLIP$_{im}$ ↑ | CLIP$_{out}$ ↑ | L1 ↓ | DINO ↑ | GPT$_{IF}$ ↑ | GPT$_{FC}$ ↑ | GPT$_{AQ}$ ↑ | GPT$_{avg}$ ↑ |
|---|---|---|---|---|---|---|---|---|
| InstructPix2Pix | 0.766 | 0.234 | 0.179 | 0.588 | 3.72 | 3.68 | 3.80 | 3.73 |
| MagicBrush | 0.806 | 0.248 | 0.148 | 0.671 | 4.52 | 4.48 | 4.38 | 4.46 |
| UltraEdit | 0.779 | 0.242 | 0.142 | 0.621 | 3.80 | 4.40 | 4.40 | 4.20 |
| ICEdit | 0.797 | 0.228 | 0.128 | 0.614 | 3.68 | 4.02 | 4.04 | 3.91 |
| IEAP(Ours) | **0.866** | **0.252** | **0.099** | **0.701** | **4.68** | **4.68** | **4.48** | **4.61** |

Table 7: Quantitative comparison results on AnyEdit Replace test set.

| Method | CLIP$_{im}$ ↑ | CLIP$_{out}$ ↑ | L1 ↓ | DINO ↑ | GPT$_{IF}$ ↑ | GPT$_{FC}$ ↑ | GPT$_{AQ}$ ↑ | GPT$_{avg}$ ↑ |
|---|---|---|---|---|---|---|---|---|
| InstructPix2Pix | 0.829 | 0.254 | 0.164 | 0.774 | 3.46 | 3.84 | 3.58 | 3.63 |
| MagicBrush | 0.831 | 0.266 | 0.156 | 0.784 | 2.96 | 4.28 | 4.28 | 3.84 |
| UltraEdit | 0.847 | 0.259 | 0.157 | 0.781 | 2.92 | 4.22 | 4.24 | 3.79 |
| ICEdit | 0.827 | 0.255 | **0.152** | 0.745 | 2.68 | 4.04 | 4.04 | 3.59 |
| IEAP(Ours) | **0.848** | **0.267** | 0.154 | **0.798** | **4.66** | **4.86** | **4.68** | **4.73** |

Table 8: Quantitative comparison results on AnyEdit Action Change test set.

| Method | CLIP$_{im}$ ↑ | CLIP$_{out}$ ↑ | L1 ↓ | DINO ↑ | GPT$_{IF}$ ↑ | GPT$_{FC}$ ↑ | GPT$_{AQ}$ ↑ | GPT$_{avg}$ ↑ |
|---|---|---|---|---|---|---|---|---|
| InstructPix2Pix | 0.881 | 0.219 | 0.127 | 0.771 | 3.82 | 4.44 | 4.36 | 4.21 |
| MagicBrush | 0.902 | 0.219 | 0.088 | 0.828 | 2.94 | 3.94 | 3.90 | 3.59 |
| UltraEdit | 0.923 | 0.211 | 0.074 | 0.867 | 3.48 | 4.40 | **4.40** | 4.09 |
| ICEdit | 0.944 | 0.213 | 0.063 | 0.868 | 3.28 | **4.64** | 4.30 | 4.07 |
| IEAP(Ours) | **0.963** | **0.223** | **0.058** | **0.903** | **3.88** | 4.44 | 4.38 | **4.23** |

Table 9: Quantitative comparison results on AnyEdit Relation test set.

| Method | CLIP$_{im}$ ↑ | CLIP$_{out}$ ↑ | L1 ↓ | DINO ↑ | GPT$_{IF}$ ↑ | GPT$_{FC}$ ↑ | GPT$_{AQ}$ ↑ | GPT$_{avg}$ ↑ |
|---|---|---|---|---|---|---|---|---|
| InstructPix2Pix | 0.831 | 0.241 | 0.124 | 0.746 | 2.94 | 3.56 | 3.62 | 3.37 |
| MagicBrush | 0.875 | 0.258 | 0.094 | 0.802 | 2.80 | 3.88 | 4.00 | 3.56 |
| UltraEdit | 0.908 | 0.262 | 0.073 | 0.889 | 3.22 | **4.38** | **4.38** | 4.00 |
| ICEdit | 0.895 | 0.253 | 0.074 | 0.841 | 3.14 | 4.28 | 4.26 | 3.89 |
| IEAP(Ours) | **0.923** | **0.263** | **0.066** | **0.921** | **4.38** | 4.32 | 4.28 | **4.32** |

Table 10: Quantitative comparison results on AnyEdit Resize test set.

| Method | CLIP$_{im}$ ↑ | CLIP$_{out}$ ↑ | L1 ↓ | DINO ↑ | GPT$_{IF}$ ↑ | GPT$_{FC}$ ↑ | GPT$_{AQ}$ ↑ | GPT$_{avg}$ ↑ |
|---|---|---|---|---|---|---|---|---|
| InstructPix2Pix | 0.815 | 0.280 | 0.139 | 0.744 | 3.60 | 4.08 | 3.92 | 3.87 |
| MagicBrush | 0.852 | **0.294** | 0.094 | 0.815 | 3.96 | 4.32 | 3.98 | 4.09 |
| UltraEdit | 0.857 | 0.277 | **0.068** | **0.845** | 4.04 | 4.62 | 4.42 | 4.36 |
| ICEdit | 0.847 | 0.273 | 0.085 | 0.808 | 4.04 | 4.42 | 4.16 | 4.21 |
| IEAP(Ours) | **0.886** | 0.285 | 0.082 | 0.833 | **4.06** | **4.72** | **4.80** | **4.53** |

Table 11: Quantitative comparison results on AnyEdit Appearance test set.

| Method | CLIP$_{im}$ ↑ | CLIP$_{out}$ ↑ | L1 ↓ | DINO ↑ | GPT$_{IF}$ ↑ | GPT$_{FC}$ ↑ | GPT$_{AQ}$ ↑ | GPT$_{avg}$ ↑ |
|---|---|---|---|---|---|---|---|---|
| InstructPix2Pix | 0.725 | 0.224 | 0.216 | 0.582 | 3.40 | 3.60 | 3.44 | 3.48 |
| MagicBrush | 0.746 | 0.230 | 0.228 | 0.567 | 4.58 | 4.38 | 4.46 | 4.47 |
| UltraEdit | 0.796 | **0.257** | 0.169 | 0.747 | 3.48 | 4.36 | 3.14 | 3.66 |
| ICEdit | 0.799 | 0.241 | 0.166 | 0.757 | 3.04 | 4.16 | 3.88 | 3.69 |
| IEAP(Ours) | **0.801** | 0.243 | **0.165** | **0.759** | **4.74** | **4.68** | **4.70** | **4.71** |

Table 12: Quantitative comparison results on AnyEdit Background Change test set.

| Method | CLIP$_{im}$ ↑ | CLIP$_{out}$ ↑ | L1 ↓ | DINO ↑ | GPT$_{IF}$ ↑ | GPT$_{FC}$ ↑ | GPT$_{AQ}$ ↑ | GPT$_{avg}$ ↑ |
|---|---|---|---|---|---|---|---|---|
| InstructPix2Pix | 0.886 | 0.279 | 0.120 | **0.876** | 3.60 | 4.40 | 4.00 | 4.00 |
| MagicBrush | 0.898 | **0.282** | 0.087 | 0.869 | 4.20 | **4.82** | 4.62 | 4.55 |
| UltraEdit | 0.890 | 0.280 | 0.065 | 0.87 | 3.80 | 4.40 | 4.20 | 4.13 |
| ICEdit | 0.896 | 0.278 | 0.073 | 0.849 | **4.72** | 4.80 | 4.64 | **4.72** |
| IEAP(Ours) | **0.911** | 0.276 | **0.059** | 0.876 | 4.62 | 4.72 | **4.78** | 4.71 |

Table 13: Quantitative comparison results on AnyEdit Color Change test set.

| Method | CLIP$_{im}$ ↑ | L1 ↓ | DINO ↑ | GPT$_{IF}$ ↑ | GPT$_{FC}$ ↑ | GPT$_{AQ}$ ↑ | GPT$_{avg}$ ↑ |
|---|---|---|---|---|---|---|---|
| InstructPix2Pix | 0.776 | 0.068 | 0.936 | 3.74 | 4.60 | 4.30 | 4.21 |
| MagicBrush | 0.770 | 0.064 | 0.940 | 3.86 | 4.48 | 4.18 | 4.17 |
| UltraEdit | 0.699 | 0.073 | 0.907 | 3.14 | 4.10 | 3.80 | 3.68 |
| ICEdit | 0.796 | 0.065 | 0.943 | 3.16 | 4.60 | 4.30 | 4.02 |
| IEAP(Ours) | **0.882** | **0.052** | **0.945** | **4.34** | **4.72** | **4.50** | **4.52** |

Table 14: Quantitative comparison results on Expression test set.

| Method | CLIP$_{im}$ ↑ | L1 ↓ | DINO ↑ | GPT$_{IF}$ ↑ | GPT$_{FC}$ ↑ | GPT$_{AQ}$ ↑ | GPT$_{avg}$ ↑ |
|---|---|---|---|---|---|---|---|
| InstructPix2Pix | 0.746 | 0.130 | 0.549 | **4.00** | 4.18 | 4.04 | 4.07 |
| MagicBrush | 0.778 | 0.110 | 0.621 | 3.36 | 4.06 | 3.84 | 3.75 |
| UltraEdit | 0.765 | 0.086 | 0.598 | 3.34 | 4.28 | 4.04 | 3.89 |
| ICEdit | 0.787 | 0.086 | 0.616 | 3.48 | 3.92 | 3.58 | 3.66 |
| IEAP(Ours) | **0.826** | **0.055** | **0.696** | **4.08** | **4.48** | **4.18** | **4.25** |

Table 15: Quantitative comparison results on Material Change test set.

| Method | CLIP$_{im}$ ↑ | L1 ↓ | DINO ↑ | GPT$_{IF}$ ↑ | GPT$_{FC}$ ↑ | GPT$_{AQ}$ ↑ | GPT$_{avg}$ ↑ |
|---|---|---|---|---|---|---|---|
| InstructPix2Pix | 0.710 | 0.212 | 0.463 | 3.56 | 4.32 | 3.94 | 3.94 |
| MagicBrush | 0.692 | 0.214 | 0.440 | 3.12 | 4.64 | 4.00 | 3.92 |
| UltraEdit | 0.703 | 0.201 | 0.467 | 4.02 | 4.8 | **4.62** | 4.48 |
| ICEdit | 0.706 | 0.219 | 0.458 | 4.04 | **4.82** | 4.36 | 4.41 |
| IEAP(Ours) | **0.922** | **0.097** | **0.915** | **4.44** | 4.64 | 4.44 | **4.51** |

Table 16: Quantitative comparison results on AnyEdit Style Change test set.

| Method | CLIP$_{im}$ ↑ | CLIP$_{out}$ ↑ | L1 ↓ | DINO ↑ | GPT$_{IF}$ ↑ | GPT$_{FC}$ ↑ | GPT$_{AQ}$ ↑ | GPT$_{avg}$ ↑ |
|---|---|---|---|---|---|---|---|---|
| InstructPix2Pix | 0.822 | 0.260 | **0.100** | 0.821 | 3.72 | 4.48 | 3.92 | 4.04 |
| MagicBrush | 0.834 | 0.266 | 0.159 | 0.791 | 3.56 | 4.64 | 3.98 | 4.06 |
| UltraEdit | 0.804 | **0.268** | 0.201 | 0.767 | 4.12 | 4.62 | 4.26 | 4.33 |
| ICEdit | 0.812 | 0.260 | 0.157 | 0.748 | 4.06 | **4.88** | **4.56** | 4.50 |
| IEAP(Ours) | **0.868** | **0.268** | 0.116 | **0.843** | **4.44** | 4.64 | 4.44 | **4.51** |

Table 17: Quantitative comparison results on AnyEdit Tone Transfer test set.

| Method | CLIP$_{im}$ ↑ | L1 ↓ | DINO ↑ | GPT$_{IF}$ ↑ | GPT$_{FC}$ ↑ | GPT$_{AQ}$ ↑ | GPT$_{avg}$ ↑ |
|---|---|---|---|---|---|---|---|
| InstructPix2Pix | 0.815 | 0.134 | 0.647 | 3.40 | 4.04 | **4.80** | 4.08 |
| MagicBrush | 0.835 | 0.081 | 0.697 | 1.82 | 3.56 | 3.50 | 2.96 |
| UltraEdit | 0.833 | 0.066 | 0.756 | 2.58 | 4.02 | 4.02 | 3.54 |
| ICEdit | 0.906 | **0.042** | **0.842** | 2.98 | 4.40 | 3.40 | 3.59 |
| IEAP(Ours) | **0.908** | 0.056 | 0.794 | **3.42** | **4.48** | 4.46 | **4.12** |

Table 18: Quantitative comparison results on AnyEdit Counting test set.

| Method | CLIP$_{im}$ ↑ | L1 ↓ | DINO ↑ | GPT$_{IF}$ ↑ | GPT$_{FC}$ ↑ | GPT$_{AQ}$ ↑ | GPT$_{avg}$ ↑ |
|---|---|---|---|---|---|---|---|
| InstructPix2Pix | 0.773 | 0.208 | 0.581 | 3.46 | 4.18 | 4.08 | 3.91 |
| MagicBrush | 0.806 | 0.174 | 0.631 | 2.98 | 3.88 | 4.04 | 3.63 |
| UltraEdit | 0.825 | **0.167** | **0.669** | 2.82 | 4.38 | 4.38 | 3.86 |
| ICEdit | 0.806 | 0.171 | 0.629 | 3.56 | 4.16 | 4.06 | 3.93 |
| IEAP(Ours) | **0.833** | 0.169 | 0.662 | **3.88** | **4.44** | **4.52** | **4.28** |

Table 19: Quantitative comparison results on AnyEdit Implicit Change test set.

| Method | CLIP$_{im}$ ↑ | L1 ↓ | DINO ↑ | GPT$_{IF}$ ↑ | GPT$_{FC}$ ↑ | GPT$_{AQ}$ ↑ | GPT$_{avg}$ ↑ |
|---|---|---|---|---|---|---|---|
| InstructPix2Pix | 0.887 | 0.111 | 0.858 | **4.30** | 4.50 | 4.30 | **4.37** |
| MagicBrush | 0.900 | 0.100 | 0.874 | 4.12 | 4.36 | **4.54** | 4.34 |
| UltraEdit | 0.922 | **0.077** | 0.911 | 3.24 | 4.4 | 4.36 | 4.00 |
| ICEdit | 0.898 | 0.079 | 0.864 | 4.16 | 4.46 | 4.20 | 4.27 |
| IEAP(Ours) | **0.938** | 0.084 | **0.925** | 4.18 | **4.56** | 4.38 | **4.37** |

Table 20: Quantitative comparison results on AnyEdit Move test set.

| Method | CLIP$_{im}$ ↑ | CLIP$_{out}$ ↑ | L1 ↓ | DINO ↑ | GPT$_{IF}$ ↑ | GPT$_{FC}$ ↑ | GPT$_{AQ}$ ↑ | GPT$_{avg}$ ↑ |
|---|---|---|---|---|---|---|---|---|
| InstructPix2Pix | 0.688 | 0.243 | 0.189 | 0.742 | 1.04 | 4.38 | 3.92 | 3.11 |
| MagicBrush | 0.680 | 0.255 | 0.156 | 0.786 | 1.02 | 4.48 | 4.10 | 3.20 |
| UltraEdit | 0.732 | 0.279 | **0.147** | **0.843** | 1.96 | 4.46 | 3.98 | 3.47 |
| ICEdit | **0.810** | **0.289** | 0.155 | 0.811 | **4.18** | 4.42 | **4.68** | **4.43** |
| IEAP(Ours) | 0.788 | 0.285 | 0.162 | 0.786 | 3.96 | **4.58** | 4.06 | 4.20 |

Table 21: Quantitative comparison results on AnyEdit Textual Change test set.

# D    More Visualization Results

In this section, we provide more visualization results, as shown below:

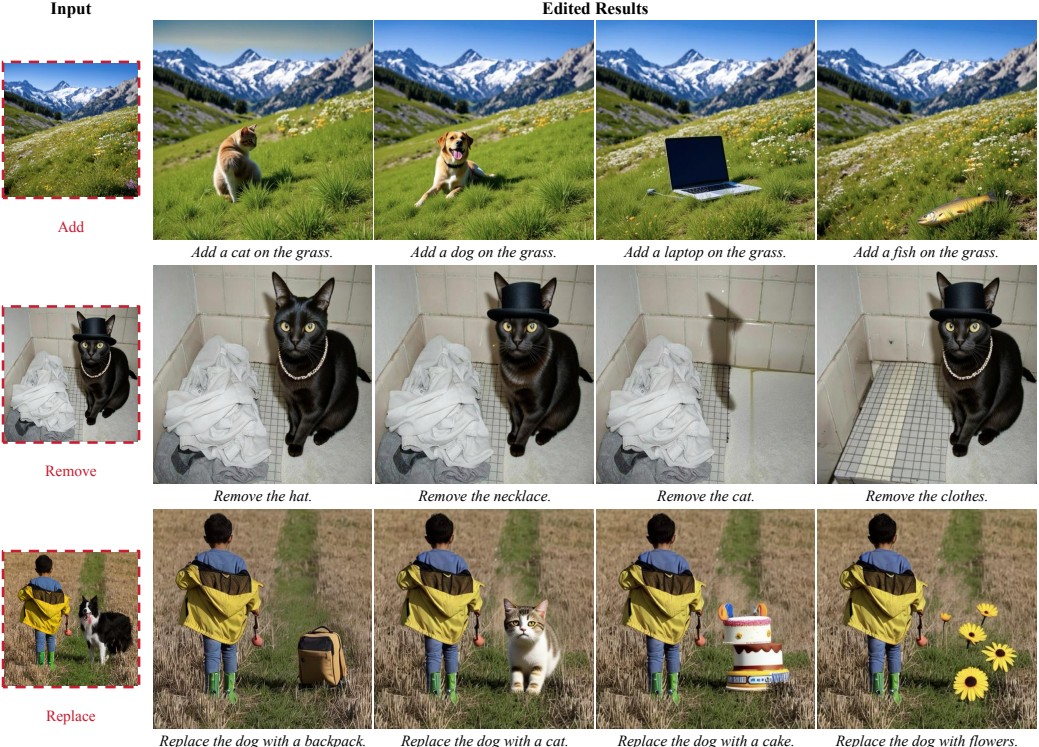

Figure 10: More Visualization Results.

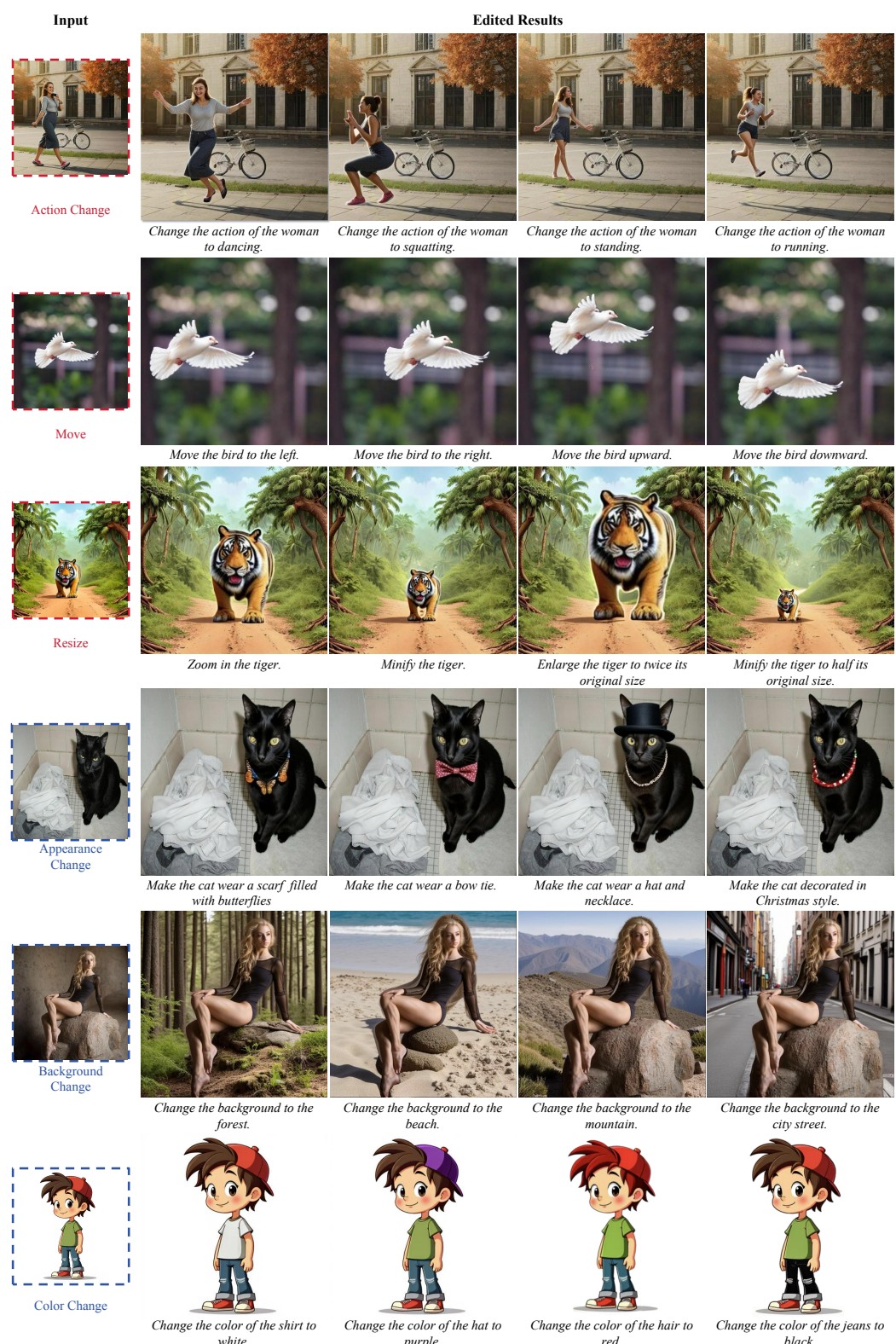

Figure 11: More Visualization Results.

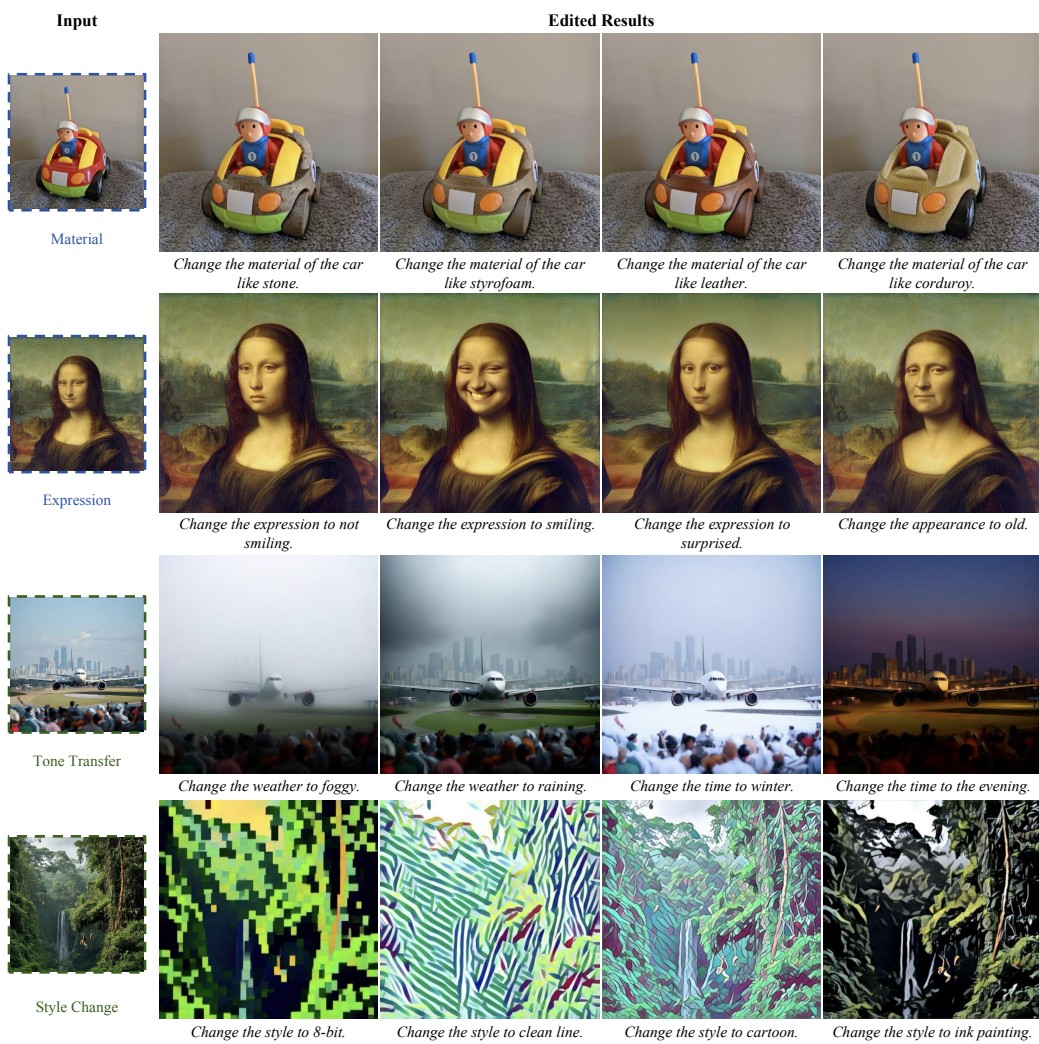

Figure 12: More Visualization Results.

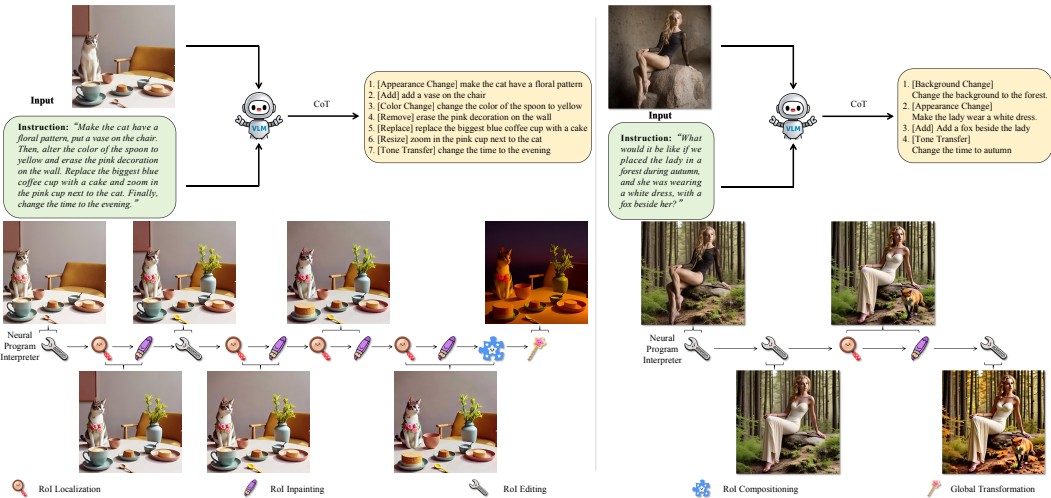

Figure 13: More Detailed Visualization Processes of the pipeline.

# E  Analysis and Discussions

## E.1  Runtime Performance Analysis

We quantified the inference latency of IEAP across different editing types on a single NVIDIA H200 GPU. For local attribute editing and overall content editing, only one atomic diffusion-based step is required. In contrast, local semantic editing typically involves multiple atomic operations, depending on the specific editing intent:

$$
\begin{aligned}
\text{Add/Replace:} &\quad 1 \times \text{RoI Localization} + 1 \times \text{Diffusion} + 2 \times \text{LLM Response}, \\
\text{Remove:} &\quad 1 \times \text{RoI Localization} + 1 \times \text{Diffusion} + 1 \times \text{LLM Response}, \\
\text{Action Change:} &\quad 2 \times \text{RoI Localization} + 3 \times \text{Diffusion} + 2 \times \text{LLM Response}, \\
\text{Move/Resize:} &\quad 1 \times \text{RoI Localization} + 2 \times \text{Diffusion} + 2 \times \text{LLM Response}.
\end{aligned}
$$

Empirical measurements show that each diffusion step takes approximately $5\,\text{s}$ when using FLUX.1-dev and around $4\,\text{s}$ with FLUX.1-schnell. The LLM response incurs about $1\,\text{s}$ latency, while RoI localization requires roughly $2\,\text{s}$. Based on these estimates, the end-to-end latencies for representative operations are approximately: Add/Replace: 8–9 s; Remove: 7–8 s; Action Change: 18–20 s; Move/Resize: 12–14 s; Other operations: typically 4–5 s.

On the AnyEdit test set, which includes 16 subtypes of editing, our method achieves a weighted average latency of approximately $7\,\text{s}$ per edit. For comparison, the end-to-end editing systems InstructPix2Pix, MagicBrush, UltraEdit, and ICEdit exhibit average latencies of $4\,\text{s}$, $4\,\text{s}$, $1\,\text{s}$, and $3\,\text{s}$, respectively. Although IEAP incurs slightly higher inference latency, it provides faithful, modular, and compositional control over both simple and complex multi-step edits, extending beyond the capabilities of purely end-to-end approaches.

## E.2  Limitations and Future Work

**Limitations.** Despite its strengths, IEAP exhibits several limitations in handling dynamic scenes and complex physical interactions. First, the RoI compositing may introduce geometric distortions or texture discontinuities when editing highly dynamic or non-rigid content, such as motion-blurred instances, and fluid or smoke effects. For example, in the task of "changing the cat's action to jumping," in Fig. 6, the rapid motion of fur can produce blurred regions that fail to blend naturally with the background. Second, RoI compositing struggles to simulate physically consistent lighting effects in scenes with reflective or refractive surfaces, sometimes resulting in mismatched shadow directions and illumination conflicts between edited objects and their environments. For example, in the task of "change the action of the woman to dancing," in Fig. 4, the shadows before and after editing remain the same, but the action of the woman has changed, so it is unnatural. Third, the DiT-based architecture and multi-stage atomic operations incur substantial inference latency for $5\,\text{s}$ to $9\,\text{s}$ per operation on a single H100 GPU, precluding real-time interactivity in applications such as AR/VR. Finally, the requirement for high-memory GPUs like NVIDIA H100 (80 GB) limits reproducibility for resource-constrained researchers, and multi-iteration editing can exacerbate image quality degradation over successive operations.

**Future Work.** As for future work, several avenues may be pursued to overcome the identified limitations. To begin with, physics-aware compositing techniques and motion-compensated inpainting could be explored to better accommodate dynamic blur and fluid effects, thereby ensuring seamless integration of non-rigid edits. Meanwhile, differentiable lighting models or neural rendering modules may be incorporated to enforce global illumination consistency, particularly in reflective and refractive contexts. On the performance front, model distillation, operation fusion, and sparse attention strategies could be investigated to reduce per-operation latency and facilitate interactive editing. To enhance accessibility, memory optimization and support for smaller-footprint architectures amenable to commodity GPUs may be implemented. Moreover, iterative refinement and error-correction mechanisms may be developed to mitigate quality degradation over successive editing steps. Furthermore, beyond still-image editing, an extension to video-based complex instruction editing could be considered, where temporal coherence and motion consistency present additional challenges and opportunities for dynamic, multi-step visual manipulation.

### E.3 Societal Impacts and Ethical Safeguards

**Positive Societal Impacts.** The proposed IEAP framework introduces a modular and interpretable approach to complex image editing, which holds significant potential to benefit a range of creative and technical domains. By decomposing high-level visual instructions into atomic operations, IEAP enables users to perform multi-step edits with enhanced precision and control. This capability is particularly valuable in digital content creation, advertising, and education, where fine-grained manipulation of visual content is often required. For example, IEAP's ability to support structurally inconsistent modifications can streamline visual storytelling workflows or facilitate the generation of accurate scientific visualizations for publications and teaching materials. Furthermore, its potential extensions to fields such as medical imaging by enabling localized enhancement of diagnostic visuals, and accessibility technology by generating descriptive visual representations for users with visual impairments, demonstrate the framework's broader societal utility and interdisciplinary relevance.

**Negative Societal Impacts and Ethical Safeguards.** Despite its benefits, IEAP's high-fidelity editing capabilities also introduce ethical risks, particularly in the domains of misinformation and privacy. The framework's precision in altering visual content could be misused for the creation of deepfakes or manipulated images intended for disinformation, identity falsification, or reputational harm. Operations such as "Remove" or "Replace" could be exploited to tamper with sensitive or private imagery, potentially infringing on individual rights.

To address these concerns, the development and deployment of IEAP adhere to strict ethical standards. Specifically, safeguards include the implementation of data filtering pipelines, such as the use of GPT-4o-filtered subsets of AnyEdit and the compliance-oriented CelebHQ-FM dataset, to reduce harmful biases and content. Additionally, the modular nature of IEAP facilitates transparency and traceability in the editing process, supporting future content provenance systems designed to detect and flag manipulated media. All these safeguards jointly contribute to ongoing efforts in AI safety and accountability.

