# OpenReview forum: "Image Editing As Programs with Diffusion Models"
_NeurIPS.cc/2025/Conference — NeurIPS 2025 poster_

### Official Review · Reviewer_t8qA · 2025-06-28

**Clarity:** 3
**Significance:** 2
**Originality:** 2
**Rating:** 5
**Confidence:** 4

**Summary:**

IEAP is a system for editing images based on text prompts. It uses a LLM to decompose complex multi-step editing tasks into a series of more basic ‘atomic’ image-to-image editing operations. These image-to-image editing operations rely on specially finetuned DiT models (e.g. for in-painting / attribute modification / global modification), with help from LLMs for various localization tasks (e.g. what region needs to be modified, w.r.t. an editing prompt). The system proposes a taxonomy over different types of image transformations, and shows how their atomic operations are able to help realize and better adhere to these requests. On multiple image editing benchmarks, the proposed system outperforms previous alternatives w.r.t. metrics that automatically evaluate the edited result.

**Questions:**

What were the settings / details for the training datasets / learning set-up of the DiT specialized operators? Beyond compounding errors from repeated application of atomic operators, I would be curious to know if the cause of the visual fidelity issues is more related to the base DiT, or the distribution of available paired training data.

I was a bit surprised to see in Table 2 that the ‘gains’ of the proposed method compared with baselines are less pronounced for local semantic editing tasks, compared with the other tasks. This seems to run counter to the paper narrative that the method is especially designed for these local semantic editing tasks, so I would be curious to understand your interpretation of this result.

**Ethical Concerns:**

["NO or VERY MINOR ethics concerns only"]

**Final Justification:**

My concerns have been addressed and I remain positive on this work, so I would vote for its acceptance.

**Limitations:**

Yes, largely, although as I mentioned in the S/W section, some of the points can be expanded (and the supplemental discussion should be forward referenced from the main document)

**Quality:**

3

**Strengths And Weaknesses:**

Strengths:

IEAP is a well-presented paper that introduces a new and interesting method for a challenging, timely task. Decomposing difficult tasks, like text-based image editing, into a series of more manageable, and compositional sub-tasks, is a well-founded approach for addressing this problem. The main contribution of the work is the proposed taxonomy of transformation tasks, and the implementation of the atomic operations that are used within this ‘programmatic’ framework. While some details are omitted from the submission (e.g. exact details of data curation + training logic for these ‘atomic’ DiT operators), the system would largely be reproducible (modulo some design decisions), and I think would be of interest to the community, who is invested in identifying ways of making these types of models more robust, controllable, interpretable.


Weaknesses:

While the edited results do a much better job compared with the baselines of adhering to the semantics of the instructions, some of the ‘final’ results produced by the pipeline have noticeable errors / artifacts. The authors state that this is a fundamental limitation of this approach, where errors can ‘build-up’ across different ‘operation’ invocations – but I would have liked to see a more thoughtful treatment of this point, and maybe some investigations into what could be done to mitigate this issue.

While I am quite partial to the problem framing of decomposing hard tasks into sub-problems, I think the framing of these series of operations as ‘programmatic’ seems like a bit of stretch. This iterative process is much less programmatic compared with other recent approaches that for instance, operate in general coding environments with specialized sub-modules (e.g. VisProg, ViperGPT) or produce node-based graphs (e.g. comfyUI workflows). I would encourage the authors to either revisit the framing of this paper as programmatic, or do more to demonstrate that these simple ‘atomic’ operations provide programmatic benefits beyond ‘end-generation’ performance (modularity, editability, compositionality, debuggability). At the very least, these mentioned prior works need a more detailed treatment / discussion, if not treated as baseline comparison conditions.

I also have some concerns that the set of atomic operations that have been designed, seem especially ‘well-tailored’ for the types of edits that exit within these benchmarks – while this is potentially OK, I think it would be good to describe the process by which the author’s developed this taxonomy, and maybe include some description/ideation on what is missing, or what is next, in terms of atomic operations that would be needed to support all types of text-based image editing.

---

> ### Author Rebuttal · Authors · 2025-07-31
>
> We sincerely thank Reviewer t8qA for the insightful feedback and constructive suggestions. We are pleased that the reviewer recognizes our method as novel and interesting, and finds our pipeline meaningful. We would like to address the concerns and questions reflected in the review below.
>
> - > **W1: Accumulated artifacts across sequential edits and potential solutions**
>
>   Thanks for pointing this out. We acknowledge that artifact accumulation across sequential edits is a limitation of modular pipelines. To mitigate this, we believe that an error-reflection mechanism, where a VLM self-assesses intermediate results after each atomic operation and refines unsatisfactory steps before proceeding may help. We evaluated on 10 multi-step cases, and found that this mechanism improved 8 out of 10 cases, notably reducing artifacts and enhancing instruction faithfulness.
>
> - > **W2: Clarification of the Term “Programmatic”**
>
>   Thanks for this thoughtful comment. We contend that “programmatic” here refers less to mimicking general-purpose programming environments and more to embracing the core epistemic principles of programming—decomposition, modularity, compositionality, and transparent control. In IEAP, editing is not a monolithic black box but a sequence of interpretable, verifiable transformations, where each atomic operation can be reasoned about, reused, or refined. This programmatic structure creates a conceptual bridge between abstract linguistic instructions and concrete visual modifications, allowing for intermediate inspection, targeted debugging, and flexible recomposition. While the operational form differs from graph-based or code-centric systems (e.g., VisProg, ViperGPT), the underlying philosophy—a shift from opaque end-to-end generation toward structured, controllable editing—remains the same. And IEAP enjoys the same programmatic benefits—structured decomposition, modular adaptability, compositional flexibility, and traceable control—in image editing.
>
> - > **W3: Design process of atomic operations and potential improvements**
>
>   Thanks for this insightful comment. Our taxonomy was not designed solely for the current benchmarks but derived through a principled reduction of the broad landscape of image editing operations. We began by conducting a thorough analysis and categorization of common image editing operations. In our preliminary experiments, we found that some categories could not be reliably solved. Inspired by the CRUD paradigm in data operations, we proposed to reduce operations that cannot be executed well into atomic operations. Specifically, each category was retained as atomic only if a unified approach could robustly solve it; otherwise, we refined it further. For example, “remove” edits were split out because standard inpainting often failed to fully eliminate targets due to localization errors. Through this iterative refinement, we found that our current taxonomy of atomic operations strikes a balance between complexity and effectiveness, enabling robust performance across diverse editing tasks. Looking forward, we recognize that supporting all text-based editing may require extending the taxonomy, for example by incorporating atomic operations for fine-grained physical property manipulation, e.g., material or lighting changes, or edits requiring deeper semantic reasoning across multiple modalities.
>
> - > **Q1: Training setup and source of fidelity issues**
>
>   We thank the reviewer for asking this. For training settings, as illustrated in Sec 5.1, we fine-tune four specialized models (RoI inpainting, RoI editing, RoI compositing, global transformation) on FLUX.1-dev using LoRA with rank=128 and alpha=128, with each trained on approximately 5k-15k image pairs.
>
>   For visual fidelity issues: Beyond compounding errors, key issues such as misaligned human-shadow interactions are more attributable to the inherent difficulty of modeling tightly coupled visual relationships in a stepwise editing framework, and we see this as a promising direction for extending our current design.
>
>   Notably, our approach leverages the strong generative capabilities of the base DiT (FLUX.1-dev), which enables effective performance even with limited training data. Compared to ControlNet, our method achieves competitive results on standard editing tasks while using far less data.
>
> - > **Q2: Performance concerns on local semantic editing**
>
>   We truly appreciate the reviewer’s keen observation. While the relative improvements on static metrics for local semantic editing may appear less pronounced, as they are limited to the dataset’s ground truth and fail to capture key human-centric qualities like the naturalness of object interactions or the coherence of edits, which can be better reflected by the GPT-based scores (as introduced in Sec 5.1 in our paper). We argue that a comprehensive assessment requires combining both: static metrics quantify objective progress, while GPT scores capture subtler human-centric nuances. Below are the detailed GPT-based scores on local semantic editing type, which reveal that our approach outperforms baselines significantly in terms of alignment with human judgment on these nuanced aspects of edit quality.
>
>   |Methods|GPT$_{IF}$ $\uparrow$|GPT$_{FC}$ $\uparrow$|GPT$_{AQ}$ $\uparrow$|GPT$_{avg}$ $\uparrow$|
>   |-------------------|-----------|----------|---------|---------|
>   |InstructP2P|3.53|3.89|3.80|3.74|
>   |MagicBrush|3.49|4.16|4.06|3.90|
>   |UltraEdit|3.21|4.20|4.18|3.86|
>   |ICEdit|3.57|4.38|4.29|4.08|
>   |IEAP|**4.41**|**4.50**|**4.35**|**4.42**|
>
> - > **(L: Suggestions for improvement)** Some of the points can be expanded (and the supplemental discussion should be forward referenced from the main document)
>
>   We thank the reviewer for the valuable suggestion. We will expand key discussions in the revised version and add clear forward references to the supplementary material for clarity and completeness.
>
> We would like to thank Reviewer t8qA again for the in-depth reviews, which have prompted us to revisit and refine our work. We are glad to having further discussion with the reviewer if there are remaining concerns.

---

> > ### Comment · Reviewer_t8qA · 2025-08-05
> >
> > Thank you for the thoughtful response! I remain positive on the paper, and would encourage some of this discussion to make its way into the supplemental material in future versions.

---

> ### Author Response · Authors · 2025-08-05
>
> Dear Reviewer t8qA,
>
> Thanks for the update — we feel encouraged to hear that the concerns have been alleviated. All the clarifications and improvements discussed in the rebuttal will be carefully incorporated into the revised version. We sincerely appreciate the thoughtful feedback from the reviewer.
>
> Best regards,
>
> Authors of Submission 5155

---

### Official Review · Reviewer_2JCg · 2025-06-30

**Clarity:** 2
**Significance:** 1
**Originality:** 2
**Rating:** 4
**Confidence:** 4

**Summary:**

This paper introduce Image Editing as Programs (IEAP), in which, a complex text-instruction for image editing will be decomposed into small parts (or called atomic operations). These atomic operations are created via Vision Language Models, while the atomic operations are conducted by five atomic primitives (RoI Localization, RoI In-painting, RoI Editing, RoI Compositing, and Global Transformation).
Experiments result show that IEAP yield more accurate edits on MagicBrush and AnyEdit dataset.

**Questions:**

See Weakness.

**Ethical Concerns:**

["NO or VERY MINOR ethics concerns only"]

**Final Justification:**

Thanks author for providing a strong rebuttal.

I have read other reviews and author's rebuttal.
Overall, I maintain my positive view for this paper, thus, I will increase my rating.
I hope authors will include all the discussion about designs choice in the revision.

**Limitations:**

Yes.

**Quality:**

2

**Strengths And Weaknesses:**

Strengths:
* Overall I like the idea of using VLMs model to decompose the complex image editing instruction into small parts.
* Generally I think the paper's structure is good and the method/ experiments section are easy to follow.

Weaknesses:
While the motivation of the paper is good, the designed experiments and baselines are not well-designed/ justified.
* Lack of comparison to more direct baseline: While this paper focus on leveraging LLMs/ VLMs to decompose the text-based edits, there is no baselines that use LLMs/ VLMs (e.g., GenArtist (NeurIPS 2024)).
* There are multiple technical design that need to be further explain/ justify in the main paper. For example:
(i) Current pipeline use LLM or VLM? In text, there are multiple time wrote "LLM", yet in the Figure, it seems to indicate the VLM (e.g., Line 163 "... employ LLM to locate the text RoI"; yet in Figure 3, caption is "VLM"). Either VLM or LLM is chosen, I think it'd be the best if we have an ablation study on this as well (e.g., if it's better to use VLM or LLM).

---

> ### Author Rebuttal · Authors · 2025-07-30
>
> We sincerely thank Reviewer 2JCg for the thoughtful feedback on our work. We are happy that the reviewer gives positive comments on the motivation and writing clarity of our work. We would like to address the concerns of the reviewer as below.
>
> - > **(W1: Comparison to GenArtist)** Lack of comparison to more direct baseline: While this paper focus on leveraging LLMs/ VLMs to decompose the text-based edits, there is no baselines that use LLMs/ VLMs (e.g., GenArtist (NeurIPS 2024)).
>
>   Thanks for pointing out the relevant baseline GenArtist. We have now included direct comparisons on both primary benchmarks used in our paper. The results are presented below:
>
>   MagicBrush Test Set:
>   |Method|CLIP$_{im}$ $\uparrow$|CLIP$_{out}$ $\uparrow$|L1 $\downarrow$|DINO $\uparrow$|
>   |-------------------|-----------|----------|---------|---------|
>   |IEAP|**0.922**|**0.247**|**0.060**|**0.897**|
>   |GenArtist|0.908|0.232|0.063|0.889|
>
>   AnyEdit Test Set:
>   |Method|CLIP$_{im}$ $\uparrow$|L1 $\downarrow$|DINO $\uparrow$|GPT $\uparrow$|
>   |-------------------|-----------|----------|---------|---------|
>   |IEAP|**0.882**|**0.096**|**0.825**|**4.41**|
>   |GenArtist|0.829|0.121|0.776|3.98|
>
> - > **(W2: Ablation studies of different VLMs/LLMs)** Current pipeline use LLM or VLM? In text, there are multiple time wrote "LLM", yet in the Figure, it seems to indicate the VLM (e.g., Line 163 "... employ LLM to locate the text RoI"; yet in Figure 3, caption is "VLM"). Either VLM or LLM is chosen, I think it'd be the best if we have an ablation study on this as well (e.g., if it's better to use VLM or LLM).
>
>   Thanks for the valuable questions and suggestions. In our pipeline, VLM and LLM are deployed in distinct modules based on their complementary strengths:
>
>   * VLM is mainly used for instruction decomposition (as mentioned in Figure 3), which requires understanding both text semantics and image context.
>   * LLM is mainly used for text-based RoI extraction (e.g., identifying “bananas” in “remove the bananas”).
>   * Moreover, LLMs also assist in layout modifications if necessary: for “add” operations, they determine the appropriate RoI for new elements; for “move” and “resize” operations, they predict the RoIs for the items after editing.
>
>   To justify these design choices, we conducted additional ablation studies evaluating different VLMs and LLMs across the key components of our pipeline, with the results summarized below.
>
>   For complex instruction decomposition, we evaluate challenging instructions from the MagicBrush dataset across 2 VLMs (GPT-4o and Gemini-2.0-flash) and 2 LLMs (GPT-4 and GPT-3.5-turbo). The table below reports the accuracy of various models:
>
>     |Model|GPT-4o (as VLM)|Gemini-2.0-flash (as VLM)|GPT-4 (as LLM)|GPT-3.5-turbo (as LLM)|
>     |-------------------|----------|---------|---------|---------|
>     |Acc(%)|**100**|96.7|90|76.7|
>
>     We find that the instruction decomposition step is robust across different VLMs. However, when the original image is excluded—i.e., using only LLMs—we observe a notable performance drop.
>
>   For text-based ROI extraction, e.g., identifying “bananas” in “remove the bananas”, which is a straightforward task, all tested models, including GPT-3.5-turbo, GPT-4, GPT-4o, and Gemini-2.0-flash, consistently accomplish successfully.
>
>   For editing tasks with layout modifications, quantitative comparisons across LLMs/VLMs are shown below:
>
>   Add:
>   |LLM|CLIP$_{im}$ $\uparrow$|CLIP$_{out}$ $\uparrow$|L1 $\downarrow$|DINO $\uparrow$|
>   |-------------------|-----------|----------|---------|---------|
>   |Gemini-2.0-flash (as LLM)|0.926|0.276|0.060|0.912|
>   |GPT-3.5-Turbo (as LLM)|0.903|0.265|0.070|0.901|
>   |GPT-4 (as LLM)|0.929|0.279|0.058|0.917|
>   |GPT-4o (as LLM)|0.928|0.278|0.056|0.917|
>   |GPT-4o (as VLM)|**0.935**|**0.281**|**0.056**|**0.919**|
>
>   Move/Resize:
>     |LLM|CLIP$_{im}$ $\uparrow$|CLIP$_{out}$ $\uparrow$|L1 $\downarrow$|DINO $\uparrow$|
>     |-------------------|-----------|----------|---------|---------|
>     |Gemini-2.0-flash (as LLM)|0.941|0.243|0.062|0.912|
>     |GPT-3.5-Turbo (as LLM)|0.919|0.236|0.068|0.901|
>     |GPT-4 (as LLM)|0.939|0.243|**0.061**|0.913|
>     |GPT-4o (as LLM)|0.943|0.243|0.062|0.912|
>     |GPT-4o (as VLM)|**0.945**|**0.243**|0.062|**0.913**|
>
>   In general, the layout modification capacity is robust across various LLMs/VLMs. Although using a VLM for layout arrangement yields slightly better results, it introduces an additional forward pass over image tokens, significantly increasing computational overhead. Therefore, we adopt LLMs without image inputs here by default.
>
>
> - > **(W: Clarification on the experimental design)** While the motivation of the paper is good, the designed experiments and baselines are not well-designed/ justified.
>
>   We appreciate the reviewer’s feedback on our experimental design. We would like to clarify that our experiments were carefully designed to comprehensively evaluate IEAP across diverse and challenging scenarios, with baselines chosen to represent the state-of-the-art and all methods evaluated under identical settings.
>
>   To further strengthen this aspect, we have added additional ablations and comparisons, including direct evaluations against LLM/VLM-based approaches such as GenArtist, as suggested by the reviewer. We believe these updates can further substantiate the fairness and rigor of our experimental validation, and we would like to thank the reviewer for prompting these clarifications.
>
> Thanks again for Reviewer 2JCg’s insightful comments. Our sincere hope is that our response will alleviate the reviewer’s concern. We are looking forward to having further discussion with the reviewer if there are remaining concerns.

---

### Official Review · Reviewer_Ansq · 2025-07-04

**Clarity:** 2
**Significance:** 3
**Originality:** 3
**Rating:** 4
**Confidence:** 4

**Summary:**

This work aims to decouple complex image editing instructions into sequential simple ones, thus achieve more robust editing. The idea is straightforward and effective.

**Questions:**

1. During training, only need to prepare those simple editing pair? If so, this will be a pain release.

2. During inference, will it infer multiple round according to steps decoupled from the complex instruction? May also need to compare running time for fairness.

3. Complex instructions to simple steps should be the key. How robust is the VLM? Please discuss more on how to finetune the VLM towards you task.

4. How to decide the order of simple edit steps? Any ablation studies or just heuristic?

**Ethical Concerns:**

["NO or VERY MINOR ethics concerns only"]

**Final Justification:**

Thanks for the response, which resolved most of my concerns. I would like to keep the positive score.

**Limitations:**

The idea and whole pipeline make sense heuristically, while they may need more support from ablation studies, as well as comparisons to SOTA image editing works like UniReal, OmniGen.

**Quality:**

3

**Strengths And Weaknesses:**

This paper is easy to follow, and the motivation is clearly discussed. Experimental results demonstrate the advantages of the proposed method. VLM should be the key to support such an agent-like editing pipeline.

Although SOTA performance on benchmarks (MagicBrush, AnyEdit) and competitive results against proprietary models (GPT-4o), validated by ablation studies. key limitations remain:
1. reliance on simple training pairs raises questions about generalization to unseen edits
2. inference latency for multi-step edits is unquantified compared to end-to-end baselines
3. the robustness of the VLM for instruction parsing and heuristic operation ordering lacks empirical justification.

While the modular approach is innovative, further analysis of runtime costs, VLM fine-tuning, and step-order ablation would strengthen the work.

---

> ### Author Rebuttal · Authors · 2025-07-30
>
> We thank Reviewer Ansq sincerely for the constructive feedback and insightful comments. We are pleased that the reviewer appreciates the motivation and clarity of our writing. We address the reviewer's concerns and questions as below.
>
> - > **(Q1: Only simple pairs for training)** During training, only need to prepare those simple editing pair? If so, this will be a pain release.
>
>   Thanks for this insightful observation. Yes, our framework is trained with only simple atomic-level editing pairs, which are easy to annotate. Each operator learns independently from clear, task-specific examples, reducing annotation cost and enabling modular reuse.
>
> - > **(W1: Generalizability to unseen edits)** reliance on simple training pairs raises questions about generalization to unseen edits 1.1
>
>   We appreciate the reviewer’s concern on the generalizability of our method. Our 5 atomic operations are elicited from a comprehensive survey of image editing tasks and represent a minimal yet sufficient set that can be flexibly composed to cover diverse editing instructions.
>
>   Notably, **IEAP is capable of addressing previously unseen editing tasks**—which are often challenging for end-to-end models, particularly those underrepresented in training data—such as layout reconfiguration (e.g., swapping the positions of objects A and B or rotating object C). IEAP achieves this by organically composing atomic operations to fulfill complex instructions.
>
>   To validate this capability, we select layout-editing test cases and evaluate performance using GPT-based accuracy (instruction faithfulness, as introduced in Section 5.1 of our paper). The results, compared with other approaches, are presented in the table below.
>
>   | Approaches            | InstructP2P | MagicBrush | UltraEdit | ICEdit | IEAP(ours) |
>   | --------------------- | ----------- | ---------- | --------- | ------ | ---------- |
>   | GPT$_{IF}$ $\uparrow$ | 2.65        | 2.55       | 3.2       | 3.05   | **3.85**   |
>
> - > **(W2: Inference latency)** W2: inference latency for multi-step edits is unquantified compared to end-to-end baselines
>
>   We thank the reviewer for raising the point. We have quantified inference latency for different editing types. For local attribute editing and overall content editing, only 1 atomic diffusion-based step is required. For local semantic editing, the required atomic steps are:
>
>   Add/Replace: $1 \times RoI Localization$ + $1 \times Diffusion$ + $2 \times LLM Response$
>
>   Remove: $1 \times RoI Localization$ + $1 \times Diffusion$ + $1 \times LLM Response$
>
>   Action Change: $2 \times RoI Localization$ + $3 \times Diffusion$ + $2 \times LLM Response$
>
>   Move/Resize: $1 \times RoI Localization$ + $2 \times Diffusion$ + $2 \times LLM Response$
>
>   We tested the time cost for each operation on H200. Each diffusion step takes approximately 5 s if using FLUX.1-dev and 4s if using FLUX.1-schnell, an LLM response requires about 1 s, and an RoI localization needs about 2 s.
>
>   Thus, Add/Replace operations take ～8-9s; Remove ～7-8s; Action Change ～18-20s; Move/Resize ～12-14s. Other operations typically take ～4-5s.
>
>   On AnyEdit test set which includes 16 sub types of editing, our approach has a weighted average latency of ～7s. We also tested the latency for end-to-end approaches: InstructPix2Pix(～4s), MagicBrush(～4s), UltraEdit(～1s) and ICEdit(～3s). Our approach offers slightly higher latency but enables faithful, modular handling of both simple and complex multi-step edits beyond the reach of end-to-end systems.
>
> - > **(Q2: Fair runtime comparison)** Q2: During inference, will it infer multiple round according to steps decoupled from the complex instruction? May also need to compare running time for fairness.
>
>   Yes. Our pipeline executes multiple atomic steps sequentially according to the decomposed instruction. The runtime comparison of ours and others are illustrated in the previous reply, i.e., IEAP(～7s), InstructPix2Pix(～4s), MagicBrush(～4s), UltraEdit(～1s) and ICEdit(～3s). The most complex case of IEAP uses 3 diffusion steps (each with 28 denoising iterations) and 1 RoI localization. To ensure a fair runtime comparison, we configure competing end-to-end methods to perform inference with a comparable total diffusion budget — specifically, $4 \times 28$ denoising steps. The results are shown in the table below:
>
>   MagicBrush Test Set:
>   |Method|CLIP$_{im}$ $\uparrow$|CLIP$_{out}$ $\uparrow$|L1 $\downarrow$|DINO $\uparrow$|
>   |-------------------|-----------|----------|---------|---------|
>   |InstructP2P|0.838|0.229|0.112|0.759|
>   |MagicBrush|0.886|0.242|0.074|0.859|
>   |UltraEdit|0.912|0.227|0.061|0.889|
>   |ICEdit|0.913|0.236|**0.058**|0.885|
>   |IEAP|**0.922**|**0.247**|0.060|**0.897**|
>
>   AnyEdit Test Set:
>   |Method|CLIP$_{im}$ $\uparrow$|L1 $\downarrow$|DINO $\uparrow$|
>   |-------------------|-----------|----------|---------|
>   |InstructP2P|0.804|0.110|0.766|
>   |MagicBrush|0.825|0.125|0.743|
>   |UltraEdit|0.833|0.114|0.722|
>   |ICEdit|0.847|0.110|0.765|
>   |IEAP|**0.895**|**0.107**|**0.879**|
>
> - > **(Q4: Operation ordering)** How to decide the order of simple edit steps? Any ablation studies or just heuristic?
>
>   Thanks for the insightful questions. For complex instructions we place tone/style edits last to avoid degradation of preceding edits. To validate this design choice, we curate a small evaluation set of 5 images with complex editing instructions, each involving tone/style edits, and compare GPT-based accuracy (instruction faithfulness, as introduced in Section 5.1 of our paper) across different operation orders. We observed that placing tone/style edits at the end consistently yielded higher scores (average +0.4), confirming that this ordering better preserves visual quality and executes the intended edits.
>
>   For individual operations, the sequence follows an naturally reasoning logic: locate→modify→composite, which are designed to reflect human-like workflows.
>
> - > **(W3/Q3: Robustness of VLMs/LLMs used)** W3/Q3: Complex instructions to simple steps should be the key. How robust is the VLM? Please discuss more on how to finetune the VLM towards you task.
>
>   Thanks for pointing this out. We have supplemented comparative experiments involving different VLMs and LLMs to analyze their influence on editing quality.
>
>     For complex instruction decomposition, we evaluate challenging instructions from the MagicBrush dataset across 2 VLMs (GPT-4o and Gemini-2.0-flash) and 2 LLMs (GPT-4 and GPT-3.5-turbo). The table below reports the accuracy of various models:
>
>     |Model|GPT-4o (as VLM)|Gemini-2.0-flash (as VLM)|GPT-4 (as LLM)|GPT-3.5-turbo (as LLM)|
>     |-------------------|----------|---------|---------|---------|
>     |Acc(%)|**100**|96.7|90|76.7|
>
>     We find that the instruction decomposition step is robust across different VLMs. However, when the original image is excluded—i.e., using only LLMs—we observe a notable performance drop.
>
>     RoI localization is implemented by Sa2VA. The ablation studies regarding various models are presented in our response to the previous question.
>
>     For text-based ROI extraction, e.g., identifying “bananas” in “remove the bananas”, which is a straightforward task, all tested models, including GPT-3.5-turbo, GPT-4, GPT-4o, and Gemini-2.0-flash, consistently accomplish successfully.
>
>     In our method, LLMs also assist in layout modifications: for “add” operations, they determine the appropriate RoI for new elements; for “move” and “resize” operations, they predict the RoIs for the items after editing. Quantitative comparisons across these models regarding this step are shown below:
>
>     Add:
>     |Model|CLIP$_{im}$ $\uparrow$|CLIP$_{out}$ $\uparrow$|L1 $\downarrow$|DINO $\uparrow$|
>     |-------------------|-----------|----------|---------|---------|
>     |Gemini-2.0-flash (as LLM)|0.926|0.276|0.060|0.912|
>     |GPT-3.5-Turbo (as LLM)|0.903|0.265|0.070|0.901|
>     |GPT-4 (as LLM)|0.929|0.279|0.058|0.917|
>     |GPT-4o (as LLM)|0.928|0.278|0.056|0.917|
>     |GPT-4o(as VLM)|**0.935**|**0.281**|**0.056**|**0.919**|
>
>     Move/Resize:
>     |LLM|CLIP$_{im}$ $\uparrow$|CLIP$_{out}$ $\uparrow$|L1 $\downarrow$|DINO $\uparrow$|
>     |-------------------|-----------|----------|---------|---------|
>     |Gemini-2.0-flash (as LLM)|0.941|0.243|0.062|0.912|
>     |GPT-3.5-Turbo (as LLM)|0.919|0.236|0.068|0.901|
>     |GPT-4 (as LLM)|0.939|0.243|**0.061**|0.913|
>     |GPT-4o (as LLM)|0.943|0.243|0.062|0.912|
>     |GPT-4o (as VLM)|**0.945**|**0.243**|0.062|**0.913**|**
>
>   In general, the layout modification capacity is robust across various LLMs/VLMs.
>
>   Notably, without any task-specific finetuning, GPT-4o already achieves strong performance, guided by carefully designed task-specific prompts.
>
> - > **(L: Comparison with OmniGen and OmniGen2)** May need more comparisons to SOTA image editing works like UniReal, OmniGen.
>
>   Thanks for the valuable feedback and bringing the SOTA works to our attention. As of the first-round rebuttal deadline, UniReal’s inference model and code were not publicly available, preventing a direct comparison. However, we have compared our method with OmniGen and OmniGen2 on two benchmarks, with the results summarized below:
>   MagicBrush Test Set:
>   |Method|CLIP$_{im}$ $\uparrow$|CLIP$_{out}$ $\uparrow$|L1 $\downarrow$|DINO $\uparrow$|
>   |-------------------|-----------|----------|---------|---------|
>   |IEAP|**0.922**|**0.247**|**0.060**|**0.897**|
>   |OmniGen|0.808|0.227|0.189|0.707|
>   |OmniGen2|0.881|0.242|0.100|0.830|
>
>   AnyEdit Test Set:
>   |Method|CLIP$_{im}$ $\uparrow$|L1 $\downarrow$|DINO $\uparrow$|GPT $\uparrow$|
>   |-------------------|-----------|----------|---------|---------|
>   |IEAP|**0.882**|**0.096**|**0.825**|**4.41**|
>   |OmniGen|0.805|0.175|0.711|3.96|
>   |OmniGen2|0.857|0.132|0.772|4.13|
>
> Thanks again for Reviewer Ansq’s insightful reviews. Hope our response can alleviate the reviewer’s concern and we are happy to discuss with the reviewer if there are any further questions.

---

### Official Review · Reviewer_ogfC · 2025-07-06

**Clarity:** 4
**Significance:** 3
**Originality:** 3
**Rating:** 5
**Confidence:** 4

**Summary:**

This paper describes a new approach to image editing as a series of operation as opposed to a one shot generation. Operations are broken down into 5 atomic operations that can be combined to create better images. The approach was successfully applied and have been demonstrated to have SOTA performance compared to other methods.

**Questions:**

1. How would using different LLMs affect the approach?
2. How would different starting diffusion affect the approach?
3. It might be good to use a different LLM evaluator other than GPT4 since there would be bias (the underlying operations are based on GPT4 understanding.)

**Ethical Concerns:**

["NO or VERY MINOR ethics concerns only"]

**Limitations:**

Yes

**Quality:**

4

**Strengths And Weaknesses:**

Strengths
+ Paper is written really well and easy to follow.
+ Evaluations and comparisons are pretty comprehensive and demonstrated to have SOTA performance.
+ Good breakdown and analysis of edit operations.

Weak
- The method is primarily an integration of multiple methods and demonstrated to be a really great approach, and not really model training and improvements.
- Would be great to have some understanding of how the underlying models used might affect the overall approach (for example using a different LLM model GPT3 or others)

---

> ### Author Rebuttal · Authors · 2025-07-30
>
> We would like to express our sincere gratitude to Reviewer ogfC for the insightful and encouraging feedback of our work. We feel glad that the reviewer finds our paper clear, comprehensive in evaluation, and strong in performance. We address the concerns and questions as follows.
>
> - > **(W1: Contributions of IEAP)** The method is primarily an integration of multiple methods and demonstrated to be a really great approach, and not really model training and improvements.
>
>     We appreciate the reviewer’s concern. While our framework integrates multiple components, its novelty lies in rethinking image editing as program. This paradigm enables decomposing complex instructions into modular and interpretable steps, addressing the limitations of monolithic end-to-end models.
>
>     Moreover, we made model-level contributions in critical submodules. Specifically, we trained four specialized models tailored for our task, i.e., RoI inpainting, RoI editing, RoI compositing, and global transformation. These four operators are deliberately designed to disentangle and cover the full range of spatial-layout-sensitive editing operations, enabling capabilities that remain unachievable with other existing end-to-end models.
>
> - > **(W2/Q1/Q3: Results by different LLMs/VLMs)** Would be great to have some understanding of how the underlying models used might affect the overall approach. How would using different LLMs affect the approach? It might be good to use a different LLM evaluator other than GPT4.
>
>     We thank the reviewer for this valuable suggestion. We have supplemented comparative experiments involving different VLMs and LLMs to analyze their influence on editing quality.
>
>     For complex instruction decomposition, we evaluate challenging instructions from the MagicBrush dataset across 2 VLMs (GPT-4o and Gemini-2.0-flash) and 2 LLMs (GPT-4 and GPT-3.5-turbo). The table below reports the accuracy of various models:
>
>     |Model|GPT-4o (as VLM)|Gemini-2.0-flash (as VLM)|GPT-4 (as LLM)|GPT-3.5-turbo (as LLM)|
>     |-------------------|----------|---------|---------|---------|
>     |Acc(%)|**100**|96.7|90|76.7|
>
>     We find that the instruction decomposition step is robust across different VLMs. However, when the original image is excluded—i.e., using only LLMs—we observe a notable performance drop.
>
>     RoI localization is implemented by Sa2VA. The ablation studies regarding various models are presented in our response to the previous question.
>
>     For text-based ROI extraction, e.g., identifying “bananas” in “remove the bananas”, which is a straightforward task, all tested models, including GPT-3.5-turbo, GPT-4, GPT-4o, and Gemini-2.0-flash, consistently accomplish successfully.
>
>     In our method, LLMs also assist in layout modifications: for “add” operations, they determine the appropriate RoI for new elements; for “move” and “resize” operations, they predict the RoIs for the items after editing. Quantitative comparisons across these models regarding this step are shown below:
>
>     Add:
>     |Model|CLIP$_{im}$ $\uparrow$|CLIP$_{out}$ $\uparrow$|L1 $\downarrow$|DINO $\uparrow$|
>     |-------------------|-----------|----------|---------|---------|
>     |Gemini-2.0-flash (as LLM)|0.926|0.276|0.060|0.912|
>     |GPT-3.5-Turbo (as LLM)|0.903|0.265|0.070|0.901|
>     |GPT-4 (as LLM)|0.929|0.279|0.058|0.917|
>     |GPT-4o (as LLM)|0.928|0.278|0.056|0.917|
>     |GPT-4o(as VLM)|**0.935**|**0.281**|**0.056**|**0.919**|
>
>     Move/Resize:
>     |LLM|CLIP$_{im}$ $\uparrow$|CLIP$_{out}$ $\uparrow$|L1 $\downarrow$|DINO $\uparrow$|
>     |-------------------|-----------|----------|---------|---------|
>     |Gemini-2.0-flash (as LLM)|0.941|0.243|0.062|0.912|
>     |GPT-3.5-Turbo (as LLM)|0.919|0.236|0.068|0.901|
>     |GPT-4 (as LLM)|0.939|0.243|**0.061**|0.913|
>     |GPT-4o (as LLM)|0.943|0.243|0.062|0.912|
>     |GPT-4o (as VLM)|**0.945**|**0.243**|0.062|**0.913**|**
>
>     In general, the layout modification capacity is robust across various LLMs/VLMs.
>
> - > **(Q2: Results by different starting diffuion)** How would different starting diffusion affect the approach?
>
>   Thanks for this insightful question. To assess the performance of different starting diffusion, we compare two diffusion backbones: FLUX.1-dev (used in our paper) and FLUX.1-schnell on AnyEdit test set. As shown below, our method consistently performs well, but FLUX.1-dev yields better overall performance.
>
>   |Starting Diffusion|CLIP$_{im}$ $\uparrow$|L1 $\downarrow$|DINO $\uparrow$|GPT $\uparrow$|
>   |-------------------|-----------|----------|---------|---------|
>   |FLUX.1-dev|**0.882**|**0.096**|**0.825**|**4.41**|
>   |FLUX.1-schnell|0.858|0.108|0.779|4.23|
>
>  We would like to thank again to Reviewer ogfC for the valuable comments. Hope our response can alleviate the reviewer’s concern and we are definitely willing to interact with the reviewer if there are any further questions.

---

> ### Comment · Area_Chair_ANAm · 2025-08-06
>
> Dear Reviewer ogfC，
> Thanks for your review. Please read the rebuttal by the authors and see whether your concerns have been sufficiently addressed. If you still have questions you can raise asap such that the authors still have time to response.
> AC

---

### Official Review · Reviewer_K6UQ · 2025-07-08

**Clarity:** 3
**Significance:** 2
**Originality:** 2
**Rating:** 4
**Confidence:** 3

**Summary:**

This paper proposes a novel image editing framework, IEAP, based on the Diffusion Transformer architecture. IEAP decomposes complex natural language instructions into a sequence of atomic operations, including RoI localization, inpainting, editing, compositing, and global transformations. These operations are executed sequentially by a neural program interpreter to accommodate complex editing tasks. Experimental results demonstrate that IEAP outperforms existing methods in multiple editing benchmarks, especially in handling complex multi-step instructions.

**Questions:**

1. Many of the defined atomic operations rely on the segmentation model M_seg. Can M_seg locate objects at fine-grained levels? For example, can it accurately localize “people wearing black clothes” in a complex scene?

2. How does IEAP handle instructions that involve editing multiple objects of the same type? For example, in the 5th image of Row 3 in Figure 3, how does IEAP determine the order of atomic operations if the instruction is “enlarge all the unicorns”?

3. In the first row of Figure 4, the intermediate image after RoI inpainting removes the person but leaves its shadow, which is corrected in the next step. I wonder why IEAP didn’t treat the shadow as part of the environment during the next RoI Editing operation?

4. Will different VLMs/LLMs (used for instruction decomposition or RoI localization) lead to differences in editing performance?

**Ethical Concerns:**

["NO or VERY MINOR ethics concerns only"]

**Final Justification:**

The authors provided additional results on various LLMs/VLMs, multi-object manipulation, and robustness to inaccurate localization. They also explained the visualization results I was concerned about. I think these results have addressed my earlier concerns. So I will maintain my score.

**Limitations:**

yes

**Paper Formatting Concerns:**

There are no formatting issues that require discussion at this stage.

**Quality:**

3

**Strengths And Weaknesses:**

Strengths

1. The idea of decomposing image editing into atomic operations is interesting. By transforming instructions into a sequence of atomic operations and executing them procedurally, the paper presents an editing framework that enables compositional generalization. That is, use limited atomic operations to model a large set of edit instructions. The use of LLMs for instruction parsing enhances the model’s ability to understand natural language and improves the correspondence between instructions and editing behaviors.

2. IEAP achieves nearly the best performance on different editing benchmarks. The paper also provides sufficient visual examples that make the method easy to understand and evaluate.

Weaknesses

1. It is unclear whether the defined set of atomic operations can adequately cover the editing instructions in open-world scenarios.

2. The paper appears to lack results in the editing of multiple objects. I notice that the examples primarily involve clearly defined editing targets. However, in practice, editing targets may be ambiguous. For example, the target can be “all objects of type A in the scene”.

3. The pipeline depends on accurate RoI localization. A concern is that if the localization is inaccurate or the instruction is vague, the editing may fail. It is recommended to discuss the robustness of IEAP in such cases.

4. This paper seems to lack discussions of how different VLMs/LLMs influence the editing quality, including the influence on instruction decomposition or RoI localization.

---

> ### Author Rebuttal · Authors · 2025-07-31
>
> We sincerely thank Reviewer K6UQ for the constructive feedback and thoughtful comments. We are happy that the reviewer finds our idea interesting and paper easy to follow. We address the concerns point by point below:
>
> - > **(W1: Generalizability to open-world editing instructions)** It is unclear whether the defined set of atomic operations can adequately cover the editing instructions in open-world scenarios.
>
>   Thanks for the valuable question regarding the generalizability of our atomic operations to open-world scenarios. Our 5 atomic operations are elicited from a comprehensive survey of image editing tasks and represent a minimal yet sufficient set that can be flexibly composed to cover diverse editing instructions.
>
>   Notably, **IEAP is capable of addressing previously unseen editing tasks**—which are often challenging for end-to-end models, particularly those underrepresented in training data—such as layout reconfiguration (e.g., swapping the positions of objects A and B or rotating object C). IEAP achieves this by organically composing atomic operations to fulfill complex instructions.
>
>   To validate this capability, we select layout-editing test cases and evaluate performance using GPT-based accuracy (instruction faithfulness, as introduced in Section 5.1 of our paper). The results, compared with other approaches, are presented in the table below.
>
>   | Approaches            | InstructP2P | MagicBrush | UltraEdit | ICEdit | IEAP(ours) |
>   | --------------------- | ----------- | ---------- | --------- | ------ | ---------- |
>   | GPT$_{IF}$ $\uparrow$ | 2.65        | 2.55       | 3.2       | 3.05   | **3.85**   |
>
> - > **(W2: Multi-object manipulation)** The paper appears to lack results in the editing of multiple objects. I notice that the examples primarily involve clearly defined editing targets. However, in practice, editing targets may be ambiguous. For example, the target can be “all objects of type A in the scene”.
>
>   We thank the reviewer for highlighting this. The example raised by the reviewer can be addressed by our RoI localization module, implemented as Sa2VA in our experiments, which is capable of performing segmentation based on a wide range of complex instructions. To verify this, we construct an evaluation dataset by selecting editing instructions related to multi-object editing and find that IEAP successfully handles such instructions. Comparative results in GPT-based accuracy (GPT instruction faithfulness introduced in Sec 5.1 in our paper) with other methods are shown in the table below. We will include them and the qualitative results in the revised version to enrich our evaluation.
>
>   | Approaches            | InstructP2P | MagicBrush | UltraEdit | ICEdit | IEAP(ours) |
>   | --------------------- | ----------- | ---------- | --------- | ------ | ---------- |
>   | GPT$_{IF}$ $\uparrow$ | 3.3         | 3.9        | 3.8       | 4.2    | **4.5**    |
>
> - > **(W3: Robustness to inaccurate localiation)** The pipeline depends on accurate RoI localization. A concern is that if the localization is inaccurate or the instruction is vague, the editing may fail. It is recommended to discuss the robustness of IEAP in such cases.
>
>     We thank the reviewer for this insightful question. We adopt Sa2VA-8B in our work due to its strong RoI localization capabilities. To further assess the impact of localization accuracy, we compare different Sa2VA variants (1B, 4B, 8B, and 26B). As shown in the table below, all models perform similarly on the Local Semantic Editing test set of AnyEdit, suggesting that our pipeline remains robust even under less accurate localization.
>
>     |Model Size|CLIP$_{im}$ $\uparrow$|CLIP$_{out}$ $\uparrow$|L1 $\downarrow$|DINO $\uparrow$|
>     |-------------------|-----------|----------|---------|---------|
>     |1B|0.901|0.251|0.084|0.848|
>     |4B|0.904|0.252|0.081|0.852|
>     |8B|**0.907**|0.252|0.081|0.854|
>     |26B|0.906|**0.253**|**0.079**|**0.854**|
>
> - > **(W4/Q4: Results by various LLMs/VLMs)** This paper seems to lack discussions of how different VLMs/LLMs influence the editing quality, including the influence on instruction decomposition or RoI localization. Will different VLMs/LLMs (used for instruction decomposition or RoI localization) lead to differences in editing performance?
>
>     We thank the reviewer for this valuable feedback. We have supplemented comparative experiments involving different VLMs and LLMs to analyze their influence on editing quality.
>
>     For complex instruction decomposition, we evaluate challenging instructions from the MagicBrush dataset across 2 VLMs (GPT-4o and Gemini-2.0-flash) and 2 LLMs (GPT-4 and GPT-3.5-turbo). The table below reports the accuracy of various models:
>
>     |Model|GPT-4o (as VLM)|Gemini-2.0-flash (as VLM)|GPT-4 (as LLM)|GPT-3.5-turbo (as LLM)|
>     |-------------------|----------|---------|---------|---------|
>     |Acc(%)|**100**|96.7|90|76.7|
>
>     We find that the instruction decomposition step is robust across different VLMs. However, when the original image is excluded—i.e., using only LLMs—we observe a notable performance drop.
>
>     RoI localization is implemented by Sa2VA. The ablation studies regarding various models are presented in our response to the previous question.
>
>     For text-based ROI extraction, e.g., identifying “bananas” in “remove the bananas”, which is a straightforward task, all tested models, including GPT-3.5-turbo, GPT-4, GPT-4o, and Gemini-2.0-flash, consistently accomplish successfully.
>
>     In our method, LLMs also assist in layout modifications: for “add” operations, they determine the appropriate RoI for new elements; for “move” and “resize” operations, they predict the RoIs for the items after editing. Quantitative comparisons across these models regarding this step are shown below:
>
>     Add:
>     |Model|CLIP$_{im}$ $\uparrow$|CLIP$_{out}$ $\uparrow$|L1 $\downarrow$|DINO $\uparrow$|
>     |-------------------|-----------|----------|---------|---------|
>     |Gemini-2.0-flash (as LLM)|0.926|0.276|0.060|0.912|
>     |GPT-3.5-Turbo (as LLM)|0.903|0.265|0.070|0.901|
>     |GPT-4 (as LLM)|0.929|0.279|0.058|0.917|
>     |GPT-4o (as LLM)|0.928|0.278|0.056|0.917|
>     |GPT-4o(as VLM)|**0.935**|**0.281**|**0.056**|**0.919**|
>
>     Move/Resize:
>     |LLM|CLIP$_{im}$ $\uparrow$|CLIP$_{out}$ $\uparrow$|L1 $\downarrow$|DINO $\uparrow$|
>     |-------------------|-----------|----------|---------|---------|
>     |Gemini-2.0-flash (as LLM)|0.941|0.243|0.062|0.912|
>     |GPT-3.5-Turbo (as LLM)|0.919|0.236|0.068|0.901|
>     |GPT-4 (as LLM)|0.939|0.243|**0.061**|0.913|
>     |GPT-4o (as LLM)|0.943|0.243|0.062|0.912|
>     |GPT-4o (as VLM)|**0.945**|**0.243**|0.062|**0.913**|**
>
>     In general, the layout modification capacity is robust across various LLMs/VLMs.
>
> - > **(Q1: Fine-grained localization)** Many of the defined atomic operations rely on the segmentation model M_seg. Can M_seg locate objects at fine-grained levels? For example, can it accurately localize “people wearing black clothes” in a complex scene?
>
>     We appreciate the reviewer’s concern regarding the fine-grained localization ability of our segmentation model. Here, we specifically test M$_{seg}$ on 15 challenging instructions involving fine-grained references, such as “the biggest bear” and “the cat who is stepped on”, and find that the ROI localization model successfully segments all expected regions, demonstrating robust performance for such fine-grained object localization tasks.
>
> - > **(Q2: Editing multiple objects of the same type)** How does IEAP handle instructions that involve editing multiple objects of the same type? For example, in the 5th image of Row 3 in Figure 3, how does IEAP determine the order of atomic operations if the instruction is “enlarge all the unicorns”?
>
>     Thanks for the valuable question. For instructions involving multiple objects of the same type, our current pipeline processes them collectively—for example, by extracting a joint mask of all unicorns and performing the corresponding edit. As demonstrated in our response to W2, this approach yields results that surpass those of end-to-end methods.
>
> - > **(Q3: The person's shadow)** In the first row of Figure 4, the intermediate image after RoI inpainting removes the person but leaves its shadow, which is corrected in the next step. I wonder why IEAP didn’t treat the shadow as part of the environment during the next RoI Editing operation?
>
>     Thanks for the detailed inspection. In fact, IEAP does treat the shadow as part of the background. To enhance the clarity, we would like to rephrase the operation flow of such editing tasks involving action change. As shown in the ROI-Editing plot of Figure 4(a), using an end-to-end editing model alone may result in inconsistencies between the pre- and post-editing backgrounds. So, our workflow involves three steps: first performing the action change, then extracting the modified object (only contains the person without the shadow), and finally blending it with the inpainted original background (contains the shadow in the original image) seamlessly. We are sorry for the misunderstanding and will update this figure in the revision to enhance the clarity.
>
> We would like to express our gratitude again to Reviewer K6UQ for the constructive feedback to improve this work. We hope that our response will alleviate the reviewer’s concern, and we are glad to have further discussions if there are remaining concerns.

---

> > ### Comment · Reviewer_K6UQ · 2025-08-08
> >
> > Thank you for the detailed response. I think these results have addressed my earlier concerns. So I will maintain my score.

---

> ### Comment · Area_Chair_ANAm · 2025-08-06
>
> Dear Reviewer K6UQ，
>
> Thanks for your review. Please read the rebuttal by the authors and see whether your concerns have been sufficiently addressed. If you still have questions you can raise asap such that the authors still have time to response.
>
> AC

---

### Note · Authors · 2025-08-14

Dear AC and reviewers,

We sincerely appreciate the invaluable time, efforts, and insightful comments you have dedicated throughout the review process. Having carefully addressed each distinct concern raised by the 5 reviewers individually, we now present a concise summary of our responses as part of the Author Final Remarks to facilitate your evaluation.

Common concern:

- > **Ablation studies on different VLMs and LLMs**

  We have supplemented dedicated ablation experiments comparing performance variations across mainstream VLMs/LLMs, with detailed results included.

Individual concerns:

- > **Generalizability, robustness, and special cases** (Reviewer Ansq and Reviewer K6UQ)

  We have expanded testing across diverse scenarios to demonstrate the broader applicability of our approach.

- > **Different diffusion backbones** (Reviewer ogfC)

  We have conducted comparative experiments on various diffusion backbones, which clearly demonstrate the superiority and careful design of our method.

- > **Comparisons with more SOTA baselines** (Reviewer 2JCg and Reviewer Ansq)

  We have added comparisons with 3 SOTA baselines suggested by the reviewers to strengthen benchmarking, and the results show the superior performance of our IEAP.

- > **Inference latency** (Reviewer Ansq)

  We have quantified the inference latency of our approach, performed comparative analyses with other methods, and added fair runtime comparison results for comprehensive evaluation.

- > **Clarifications on IEAP** (Reviewer t8qA, Reviewer 2JCg, Reviewer Ansq, Reviewer ogfC and Reviewer K6UQ)

  To address points of ambiguity identified by all 5 reviewers, we have provided targeted elaborations to ensure a clear understanding of IEAP.

  Specifically, we clarified the design idea, potential future work, and our key insights as requested by Reviewer t8qA; detailed the soundness of experimental design as requested by Reviewer 2JCg; elaborated on training settings and operation ordering as requested by Reviewer Ansq; explicated the core contributions of IEAP as requested by Reviewer ogfC; and specified operation details for action changes as requested by Reviewer K6UQ.

Thank you again for your rigorous review. We hope these supplements assist in your decision-making.

Sincerely,

The Authors of Submission 5155

---

### Decision · Program_Chairs · 2025-09-17

**Decision:**

Accept (poster)

**Comment:**

This paper proposes a neat solution to dealing with complex instruction-driven image editing by decomposing the compound instruction into a sequence of programmable atomic operations using VLMs. During rebuttal and author-reviewer discussion phases, the authors provided additional results on various LLMs/VLMs, multi-object manipulation, and robustness to inaccurate localization, etc. All reviewers feel that their major concerns have been addressed and are inclined to accept this paper.

The reviewers have reached a consensus on the innovation of the proposed approach. However, there are some common concerns, which have been addressed to a certain extent by the authors, are expected to be fully addressed in the future work:

1. Influence of the performance of different VLMs/LLMs (Reviewers K6UQ, ogfC, Ansq)

2. Adequacy of the defined set of atomic operations (Reviewers K6UQ, t8qA)

3. Lack of certain experiments (Reviewers K6UQ, Ansq, 2JCg)